# An epigenetic switch regulates the ontogeny of AXL-positive/EGFR-TKi-resistant cells by modulating miR-335 expression

Polona Safaric Tepes[1,2†], Debjani Pal[1,3†], Trine Lindsted[1], Ingrid Ibarra[1], Amaia Lujambio[4], Vilma Jimenez Sabinina[1], Serif Senturk[1], Madison Miller[1], Navya Korimerla[1,5], Jiahao Huang[1], Lawrence Glassman[6], Paul Lee[6], David Zeltsman[6], Kevin Hyman[6], Michael Esposito[6], Gregory J Hannon[1,7], Raffaella Sordella[1,8]*

[1]Cold Spring Harbor Laboratory, Cold Spring Harbor, United States; [2]Faculty of Pharmacy University of Ljubljana, Ljubljana, Slovenia; [3]Graduate Program in Molecular and Cellular Biology, Stony Brook University, Stony Brook, New York, United States; [4]Icahn School of Medicine at Mount Sinai, Hess Center for Science and Medicine, New York, United States; [5]Graduate Program in Biomedical Engineering, Stony Brook University, New York, United States; [6]Northwell Health Long Island, Jewish Medical Center, New York, United States; [7]Cancer Research UK – Cambridge Institute, University of Cambridge, Cambridge, United Kingdom; [8]Watson School of Biological Sciences, Cold Spring Harbor Laboratory, Cold Spring Harbor, United States

**\*For correspondence:**
sordella@cshl.edu

[†]These authors contributed equally to this work

**Competing interests:** The authors declare that no competing interests exist.

**Abstract** Despite current advancements in research and therapeutics, lung cancer remains the leading cause of cancer-related mortality worldwide. This is mainly due to the resistance that patients develop against chemotherapeutic agents over the course of treatment. In the context of non-small cell lung cancers (NSCLC) harboring EGFR-oncogenic mutations, augmented levels of AXL and GAS6 have been found to drive resistance to EGFR tyrosine kinase inhibitors such as Erlotinib and Osimertinib in certain tumors with mesenchymal-like features. By studying the ontogeny of AXL-positive cells, we have identified a novel non-genetic mechanism of drug resistance based on cell-state transition. We demonstrate that AXL-positive cells are already present as a subpopulation of cancer cells in Erlotinib-naïve tumors and tumor-derived cell lines and that the expression of AXL is regulated through a stochastic mechanism centered on the epigenetic regulation of miR-335. The existence of a cell-intrinsic program through which AXL-positive/ Erlotinib-resistant cells emerge infers the need of treating tumors harboring EGFR-oncogenic mutations upfront with combinatorial treatments targeting both AXL-negative and AXL-positive cancer cells.

## Introduction

Each year, more than a million patients worldwide are diagnosed with non–small cell lung cancer (NSCLC) (*Brose et al., 2002*; *Samuels et al., 2004*; *Stephens et al., 2004*; *Haber et al., 2005*; *Bean et al., 2007*; *Pillai and Ramalingam, 2012*). In 2014, the discovery that EGFR-oncogenic mutations were present in 15–30% of NSCLC patients and that the vast majority of patients harboring such mutations are particularly sensitive to treatment with EGFR inhibitors (TKi) such as Erlotinib and

Gefitinib was a critical breakthrough (**Lynch et al., 2004**; **Paez et al., 2004**). The identification of these actionable EGFR-oncogenic mutations revolutionized the management of NSCLC tumors from a predominantly clinical-pathological to a genotype-directed classification and therapeutic approach. Yet, the success of this biomarker-based targeted therapy has been hampered by the occurrence of drug resistance. In fact, within a year of treatment with EGFR TKIs, almost all patients experience relapse (**Bell et al., 2005**).

The past 10 years have seen tremendous progress in our understanding of the multiple mechanisms that lead to acquired resistance against TKIs. Using both experimental systems and patient samples, secondary/gatekeeper mutations in *EGFR* (T790M), c-Met amplifications, *PI3K* mutations, and the acquisition of mesenchymal and small-cell lung cancer features have been identified and validated as molecular determinants of EGFR TKi resistance (**Bell et al., 2005**; **Engelman et al., 2006**; **Shaw et al., 2009**; **Yao et al., 2010**; **Shaw and Engelman, 2016**). More recently, the expression of AXL has also been reported as an additional mechanism of acquired resistance in EGFR TKi-resistant tumors with mesenchymal-like features (**Zhang et al., 2012**; **Byers et al., 2013**; **Walter et al., 2013**; **Elkabets et al., 2015**).

AXL is a member of the TAM (Tyro-AXL-Mer) receptor tyrosine kinase family. These receptors regulate a variety of cellular responses including cell survival, proliferation, motility, as well as differentiation (**Zhang et al., 2008**; **Ghosh et al., 2011**; **Ben-Batalla et al., 2013**). AXL is expressed in many embryonic tissues and participates in mesenchymal and neuronal development. In adult tissue, its expression is usually restricted to smooth muscle cells, but it has been observed to be overexpressed in several human tumors of different tissue origins (**Zhang et al., 2008**; **Ghosh et al., 2011**; **Ben-Batalla et al., 2013**).

AXL possesses an extracellular domain with two N-terminal immunoglobulin (Ig)-like domains and two fibronectin type III (FNIII) repeats that bind to the growth-arrest-specific 6 (GAS6) ligand (**O'Bryan et al., 1991**; **Mark et al., 1996**; **Nagata et al., 1996**). The binding of AXL to GAS6 – upon its paracrine or autocrine secretion – enables the trans-auto-phosphorylation of AXL's intracellular tyrosine kinase domain and, consequently, the activation of multiple downstream signaling cascades (**Braunger et al., 1997**; **Prasad et al., 2006**).

In the context of NSCLC, higher levels of AXL and GAS6 have been observed in tumors that developed resistance to Erlotinib and Osimertinib (**Zhang et al., 2012**; **Byers et al., 2013**; **Taniguchi et al., 2019**; **Chen and Riess, 2020**). In these tumors, targeting AXL by either chemical or genetic inhibition restored Erlotinib sensitivity. Alternatively, forced expression of an active AXL kinase in Erlotinib-sensitive tumor cells was sufficient to induce Erlotinib resistance (**Zhang et al., 2012**).

Despite these documented findings, the molecular mechanisms leading to the ontogeny of AXL-positive cells remains poorly understood. Unlike other receptor tyrosine kinases, no mutations or amplifications of the AXL locus have been described in AXL-positive/Erlotinib-resistant cells (**Wu et al., 2014**).

Here, we demonstrate that AXL-positive cells are already present in Erlotinib-naïve tumors and that they are generated via an epigenetic/stochastic mechanism. Consistent with this model, we found that the transition between AXL-positive and AXL-negative cells is highly plastic.

This mechanism conceptually differs from previously described models of acquired or adaptive resistance based on the acquisition of secondary mutations or drug-driven rewiring of signaling networks. The generation of AXL-positive cells is neither generated via genetic mutations nor dependent on the micro-environment or drug treatment (**Bell et al., 2005**; **Engelman et al., 2006**; **Shaw et al., 2009**; **Yao et al., 2010**; **Shaw and Engelman, 2016**). Also different from quiescent AKT1$^{low}$ cancer cells described by the Ramaswamy group, AXL-positive cells are actively dividing (**Kabraji et al., 2017**).

At the molecular level, we showed that the generation of AXL-positive cells is centered on the methylation of a specific CpG island present in the promoter of *MEST*, a gene that contains the miRNA miR-335 in its second intron. In particular, we showed that forced down-regulation of miR-335 in AXL-negative cells was sufficient to increase the expression of AXL and to induce phenotypic and molecular features that are characteristic of AXL-positive cells, such as epithelial-to-mesenchymal transition and Erlotinib resistance.

Altogether these observations define a novel mechanism that couples epigenetic/stochastic inheritance to the ontogeny of the AXL-positive/Erlotinib-resistant cells. This novel framework could

inform the development of novel cancer treatments based on the targeting of both AXL-negative and AXL-positive cell populations.

## Results

### AXL-positive cells are pre-existing in cell lines and tumors

It has been shown that when non–small cell lung cancer (NSCLC)-derived cell lines harboring EGFR-oncogenic mutations are exposed to EGFR TKis like Erlotinib, populations of AXL-positive/Erlotinib-resistant cells emerge with features similar to those observed in tumors that have developed Erlotinib treatment resistance in patients (*Zhang et al., 2012*). This is the case for the NSCLC-derived cell lines H1650-M3 and PC14. These cells are derivative of H1650 and PC9 cells, respectively, harbor EGFR -oncogenic mutations, and were previously generated by culturing the parental cells with constant high concentrations of Erlotinib (*Yao et al., 2010*).

We wondered if AXL-positive cells are present in tumors before treatment as well as in tumor-derived cell lines and whether these cells bear phenotypic and molecular similarities to the AXL-positive cells that are generated upon exposure to EGFR TKi (*Zhang et al., 2012*).

Given that AXL is a cell surface receptor, we utilized FACS-sorting analysis with an antibody that recognizes an epitope localized within the N-terminal extracellular moiety of AXL to identify and separate putative AXL-positive cells. By using the AXL-positive cell lines, H1650-M3 and PC14 as reference (*Figure 1A,B*), we observed the presence of AXL-positive cells in multiple Erlotinib-naïve cell populations (*Figure 1B–D*). The presence of these AXL-positive cells was not restricted to tumor-derived cell lines harboring EGFR-oncogenic mutations, as we observed that a similar percentage of AXL-positive cells were present also in cell lines driven for example by mutant KRAS (i.e., A549) (*Figure 1C,D*).

In tumors, the expression of AXL is often accompanied by the expression of its ligand, GAS6 resulting in the constitutive activation of AXL and its downstream signaling pathways (i.e., AKT and ERK). We found that this was the case also in the pre-existing FACS-sorted AXL-positive cells. Our RT-PCR and western blot analysis confirmed the high expression of AXL and GAS6 in these cells (*Figure 1D–F*) and indicated that AXL, as well as AKT, were constitutively phosphorylated in AXL-positive cells (*Figure 1F*).

To exclude the possibility that our observations were an artifact of our cell culture system and more importantly to test the relevance of our findings in patients, we performed similar analyses in five primary NSCLC tumors. To limit our analysis only to tumor cells, we analyzed AXL expression only in cells that were CD45$^-$, CD31$^-$, and EPCAM$^{mid/high}$. This FACS algorithm excludes bone marrow-derived cells, endothelial cells, and fibroblasts. Also in this case, we found that human primary drug-naïve tumors contained a subpopulation of cells with high expression of AXL and GAS6 (*Figure 1G,H*, *Figure 1—figure supplement 1A*).

### Pre-existing AXL-positive cells have phenotypic and molecular features of Erlotinib-resistant cells

Having shown the existence of AXL-positive cell populations in primary tumors and in tumor-derived cell lines, next, we tested whether these cells had phenotypic and molecular features of Erlotinib-resistant AXL-positive cells. We found that AXL-positive FACS-sorted cells from Erlotinib-naïve cell lines (i.e., PC9 AXL+ve) and AXL-positive cells that were generated upon Erlotinib selection (i.e., PC14) had similar sensitivity to Erlotinib treatment with IC$_{50}$ almost three times higher than parental cells (i.e., PC9) (*Figure 2A*). To further investigate the contribution of pre-existing AXL+ cells to Erlotinib resistance, we did a cell lineage tracing experiment in which drug sensitivity was assessed after that AXL+/GFP+ cells were mixed with AXL-/GFP-negative cells in the approximate equal ratio. We observed a substantial increase in the representation of the AXL+ GFP+ cells upon Erlotinib treatment (*Figure 2—figure supplement 1A–D*). Because this study was conducted in a short period (96 hr after sorting), our data further solidify our conclusion that pre-existing AXL+ cells can be the main source of Erlotinib resistance.

To account for possible differences in growing conditions, as an alternative approach, we performed a colony assay in which AXL+ and AXL– cells were mixed with different representations. We

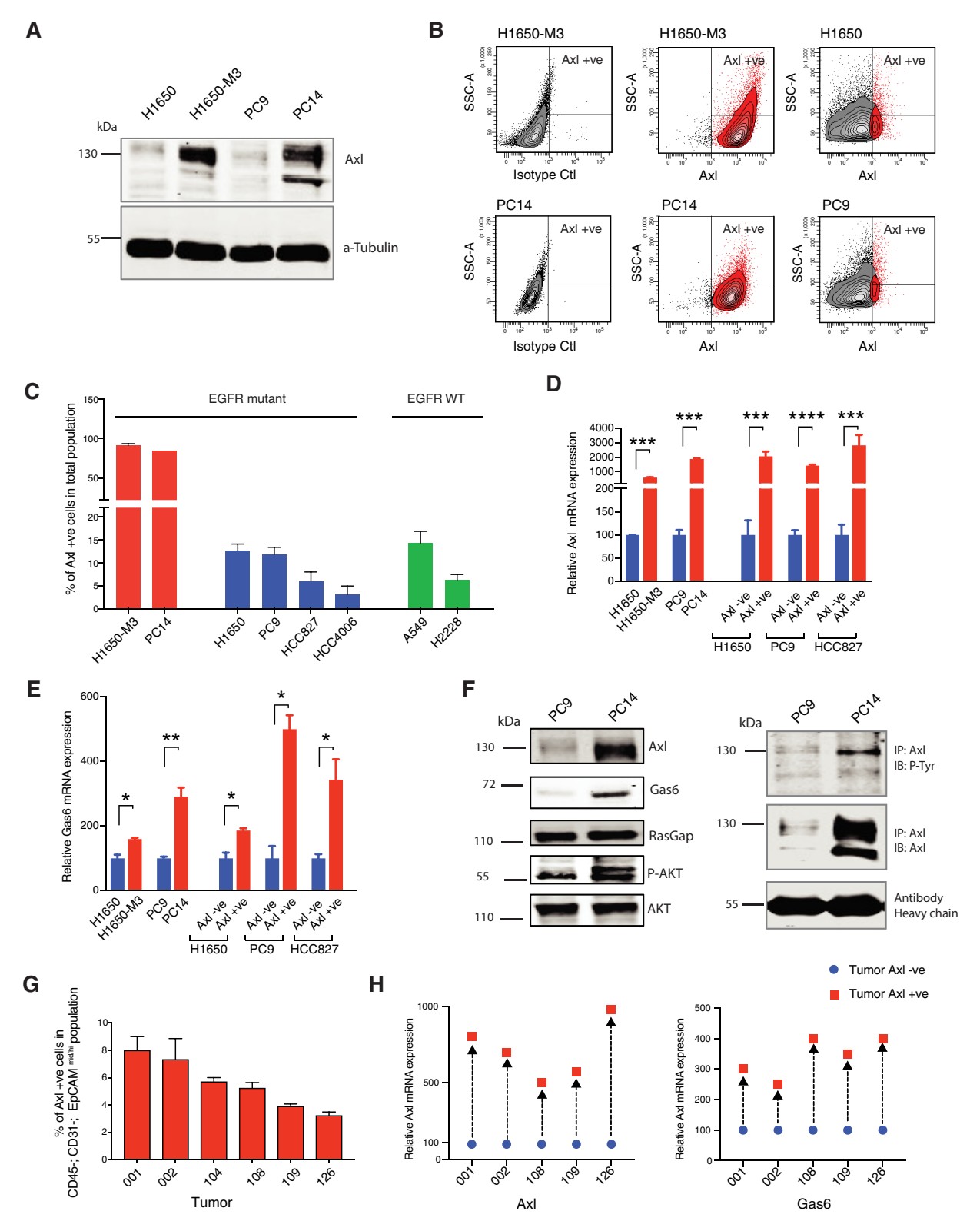

**Figure 1.** AXL-positive cells are pre-existing in cell lines and tumors. (**A**) Immunoblot analysis of AXL in AXL-positive cells (H1650-M3 and PC14) and AXL-negative cells (H1650 and PC9). α-Tubulin is used as a loading control. (**B**) Flow cytometry-based analysis of surface expression of AXL in the AXL-positive cell lines (H1650-M3 and PC14) and AXL-negative cells (H1650 and PC9). Monoclonal antibody against the N-terminal of AXL was used for the FACS analysis. Isotype control was used for identifying the AXL-negative population. (**C**) The chart represents the percentage of AXL-positive cells

*Figure 1 continued on next page*

*Figure 1 continued*

present in Erlotinib-resistant and Erlotinib-naïve cell lines. Erlotinib-resistant cell lines are indicated in red, Erlotinib naïve EGFR mutant cell lines are indicated in blue, and Erlotinib-naïve EGFR WT cell lines are indicated in green. Each bar represents mean ± SD of three replicates from two independent experiments. (D) The chart represents relative AXL mRNA expression in the indicated cell lines or cells sorted based on surface expression of AXL. Expression in AXL-positive cells was calculated relative to its expression in AXL-negative control cells. mRNA expression was quantified by SYBR-green-based RT-qPCR. Each bar represents mean ± SD of three replicates from two independent experiments (***p<0.0005, ****<0.00005, unpaired t-test). (E) The chart represents relative Gas6 mRNA expression in the indicated cell lines or cells sorted based on surface expression of AXL. Expression in AXL-positive cells was calculated relative to its expression in AXL-negative control cells. mRNA expression was quantified by SYBR-green-based RT-qPCR. Each bar represents mean ± SD of three replicates from two independent experiments (*p<0.05, **<0.005, unpaired t-test). (F) On the left panel, immunoblot analysis of AXL, GAS6, p120 RASGAP (loading control), p-AKT and AKT in AXL-negative (PC9) and AXL-positive (PC14) cells. On the right, cell extracts were immunoprecipitated with anti-AXL antibody and immunoblotted with phospho-tyrosine and AXL antibodies. Antibody heavy chain is shown as a loading control for immunoprecipitation. (G) The chart represents the percentage of AXL-positive cells in six NSCLC patient tumors. Tumor-derived single-cell suspension was stained with antibodies against CD45, CD31, EpCAM, and AXL. CD45-; CD31-; EpCAM+ cells were then FACS sorted for the AXL-positive populations. Each bar represents mean ± SD of three technical replicates. 20,000 cells were analyzed by FACS for each replicate of each sample. Schematic of the FACS sorting is presented in *Figure 1—figure supplement 1A*. Expression of *AXL* and *GAS6* genes in FACS-sorted AXL-negative (blue) and AXL-positive (red) cells from five human primary NSCLC tumors. mRNA expression was quantified by Cells to CT one-step SYBR-green-based RT-qPCR. The expression of an indicated mRNA in the AXL-positive cells was calculated relative to its expression in AXL-negative cells from the respective tumor. Each dot represents mean ± SD of three replicates.

The online version of this article includes the following figure supplement(s) for figure 1:

**Figure supplement 1.** FACS-sorting algorithm utilized to sort human tumors.

found that the presence of AXL-positive cells resulted in a significantly higher number of drug-resistant colonies, compared with only AXL-negative cells (*Figure 2—figure supplement 2A,B*).

Morphologically, the FACS-sorted AXL+ cells looked very similar to the AXL+ cells that emerged following Erlotinib treatment (PC14 and H1650-M3). All possessed the morphological and molecular features of mesenchymal cells, including loss of cobblestone shape and increased stress fibers (*Figure 2B*, *Figure 2—figure supplement 2C*) and differential expression of mesenchymal and epithelial markers (e.g., TGF-β1, TGF-β2, Slug, Twist, Vimentin, and Zeb1) (*Figure 2C*, *Figure 2—figure supplement 2D*; *Zhang et al., 2012*; *Byers et al., 2013*). These phenotypic features were driven by AXL because the inactivation of AXL in AXL-positive cells using the pharmacological inhibitor BMS-777607 resulted in the loss of the mesenchymal marker Vimentin and increased expression of E-cadherin (*Figure 2—figure supplement 2E*).

Epithelial-to-mesenchymal transition can be induced by multiple cues, including the over-expression of certain receptor tyrosine kinase receptors like AXL, c-MET, PDGFR; exposure to TGF-β1, TGF-β2; or hypoxia (*Yao et al., 2010*; *Wu et al., 2013*; *Zhang et al., 2013*; *Rankin et al., 2014*; *Elkabets et al., 2015*; *Li et al., 2015*). Hence, we wondered whether the expression of AXL was a common feature of all mesenchymal cells or if on the contrary was specific to a particular cell state. Hence, we analyzed the presence of AXL-positive cells in multiple tumor-derived cell lines and correlate their distribution with the mesenchymal status of the cells. Despite H1703, H1975, and H23 cells present with clear mesenchymal characteristics, AXL-positive cells were represented at a very low percentage in these cell lines and virtually absent in the H1703 cells (*Figure 2—figure supplement 3A–C*). Hence we concluded that while all mesenchymal cells share common characteristics such as increased stress fibers, increased motility, elongated shape, etc.; AXL-positive cells are a unique cell population with features that only partially overlap with other mesenchymal cells.

## AXL-positive cells are generated stochastically

Cancer cells are characterized by intrinsic genetic instability that can give rise to clonal cell populations with distinctive genotypic and phenotypic qualities (*Greaves and Maley, 2012*; *Barber et al., 2015*). In addition, it has been shown that intra-tumor heterogeneity could be spurred by non-genetic determinants (*Polyak and Weinberg, 2009*; *Meacham and Morrison, 2013*). In this regard, *Gupta et al., 2011* have suggested that cancer cells can oscillate stochastically among different cell states characterized by differential expression of the surface markers CD44 and CD24. More recently, the Haber group also showed that circulating tumor cells from ER+/HER− patients can be HER2− and HER2+ and readily interconvert from one state to the other within four doubling times (*Jordan et al., 2016*).

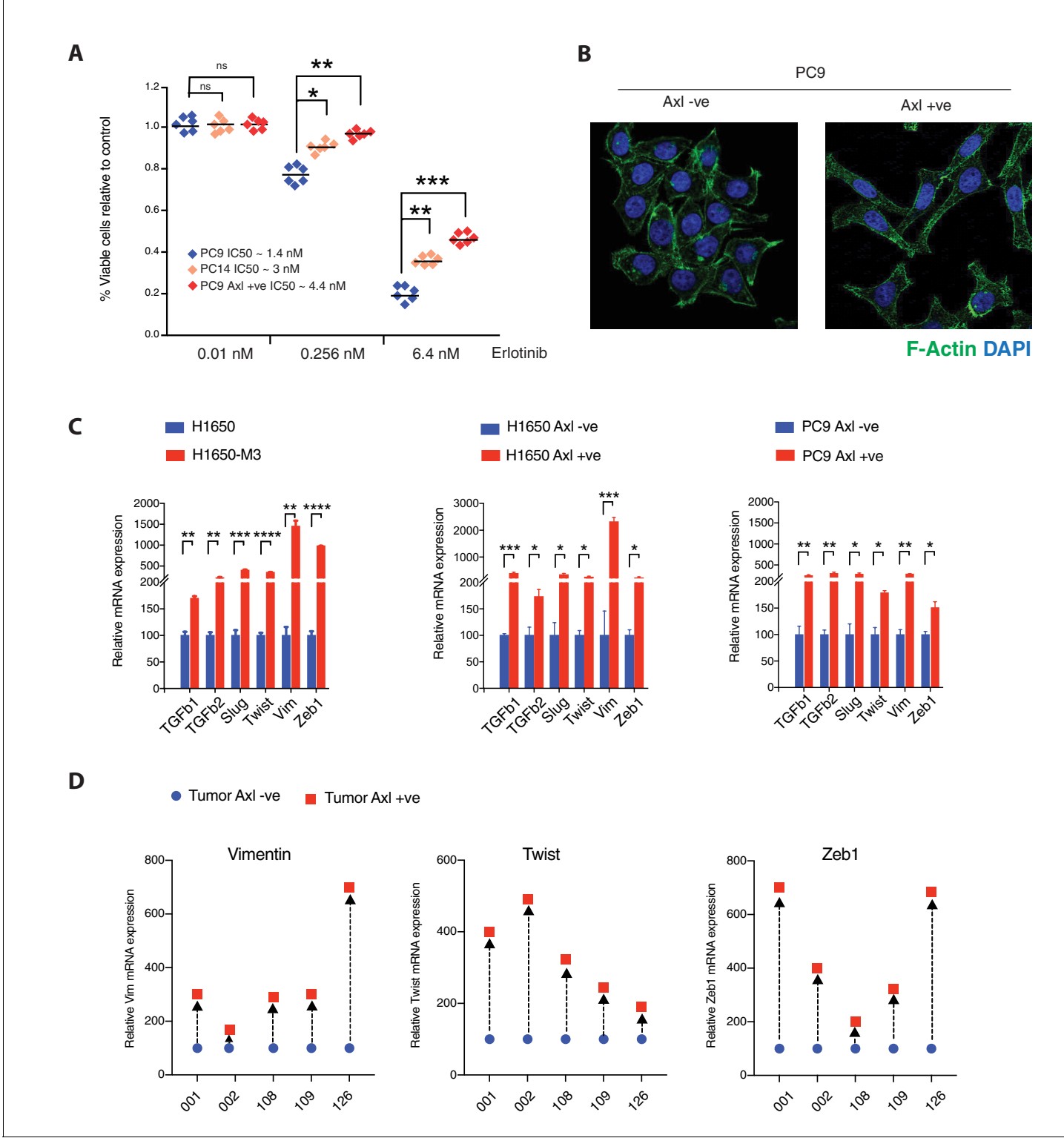

**Figure 2.** Pre-existing AXL-positive cells have characteristics of Erlotinib-resistant cells. (**A**) The chart represents the number of viable cells in PC9, PC14, and AXL-positive cells sorted from PC9 upon treatment with indicated doses of Erlotinib. Values are normalized relative to vehicle-treated cell (control). Cells were grown for 120 hr in the presence of the drug; the number of cells was estimated upon staining with the crystal violet, de-staining in 100 μl of 10% acetic acid and reading absorbance at 590 nm. Diamonds and black bars represent single-point measurements and the mean, respectively (n = 6); (*p< 0.05, **<0.005, ***<0.0005, unpaired t-test). (**B**) AXL-positive cells are characterized by mesenchymal features such as an increase in stress fibers. AXL-negative and AXL-positive cells sorted from PC9 were stained F-actin with Phalloidin (green). DAPI (blue) was used as a counter-stain. (**C**) The
*Figure 2 continued on next page*

*Figure 2 continued*

charts represent the relative expression of the indicated mesenchymal signature genes in H1650, H1650-M3, and cells sorted based on surface expression of AXL from H1650 and PC9. Expression of an indicated mRNA in the AXL-positive cells was calculated relative to its expression in AXL-negative control cells mRNA expression was quantified by SYBR-green-based RT-qPCR. Each bar represents mean ± SD of three replicates from two independent experiments (*p< 0.05, **<0.005, ***<0.0005, ****<0.00005, unpaired t-test). (D) Expression of mesenchymal signature genes *VIM*, *TWIST*, and *ZEB1* in FACS-sorted AXL-negative (blue) and AXL-positive (red) cells from five human primary NSCLC tumors. mRNA expression was quantified by Cells to CT one-step SYBR-green-based RT-qPCR. The expression of an indicated mRNA in the AXL-positive cells was calculated relative to its expression in AXL-negative cells from the respective tumor. Each dot represents mean ± SD of three replicates.

The online version of this article includes the following figure supplement(s) for figure 2:

**Figure supplement 1.** The charts represent changes in the distribution of AXL+ and AXL− cell populations at different Erlotinib concentrations.

**Figure supplement 2.** Pre-existing AXL-positive cells have characteristics of Erlotinib-resistant cells.

**Figure supplement 3.** AXL-positive cells are a unique cell population.

---

Here we tested if AXL-positive cells were generated stochastically. We reasoned that if the AXL-positive cells were generated by mutations, it would be very unlikely that these mutations would occur in synchrony. If this was the case, then we would expect the percentages of AXL-positive cells to vary across clonal cell lines derived from a single AXL-negative cell (*Figure 3A*). On the other hand, if the AXL-positive cells were generated through a stochastic event, we instead would predict the percentages of AXL-positive cells to be similar in multiple clonal cell lines derived from a single AXL-negative cell (*Figure 3B*).

To explore these two models, we derived isogenic cell lines from FACS-sorted AXL-negative H1650 and HCC827 cells; allowed them to expand; and then assessed the frequency of AXL-positive cells from four, single-cell derived clonal cell lines. We observed a very similar percentage of AXL-positive cells in the parental cells as well as in the single-cell derived clonal cell lines (*Figure 3C,D*, *Figure 3—figure supplement 1A,B*). Based on this finding, we concluded that AXL-positive cells are most likely generated from AXL-negative cells via a non-genetic, stochastic mechanism.

To further confirm this observation and to improve our understanding of the cell-state plasticity of AXL-positive and AXL-negative cells, we sorted pure AXL-positive and AXL-negative cells from the H1650 cell line and analyzed the distribution of AXL-positive and AXL-negative progeny of cells over time (*Figure 3E*). We found that within 3 weeks, the AXL-negative cells could regenerate cell populations with the same percentage of AXL-positive and AXL-negative cells as the parental cell line. Interestingly, we observed that even though the AXL-positive cells took a longer time to do so (18 weeks), they too were able to regenerate a progeny population with the same percentages of AXL-positive and AXL-negative cells as present in the parental cell line. To exclude the possibility that this finding was the result of competition among clones driven by genetic mutations, we repeated the same experiments using a single-cell derived cell line (e.g., H1650- clone 2). In this case, a nearly identical trend was recapitulated (*Figure 3F*). miRNA profiling of AXL-positive cells revealed a unique miRNA signature.

Among the many possible regulators of cell-state plasticity, we sought to investigate whether microRNAs (miRNAs) were involved in modulating the ontogeny of AXL-positive cells (*Garzon et al., 2010*).

miRNAs are small (~22 nt) non-coding RNAs constituting a novel class of gene regulators that post-transcriptionally repress gene expression by initiating the degradation or blocking translation of target mRNAs (*Lau et al., 2001*; *Lee and Ambros, 2001*; *Ambros et al., 2003*). More than 1000 unique, mature miRNAs have been identified in the human genome (*Griffiths-Jones, 2004*), and each may regulate up to 200 mRNAs (*Lewis et al., 2003*; *Betel et al., 2008*). It is estimated that roughly 30% of all human gene transcripts are targeted by miRNAs, implicating them in the regulation of virtually all cellular processes.

We generated miRNA expression profiles from the AXL-positive H1650-M3 and parental AXL-negative H1650 cells by constructing small RNA libraries. These libraries were deep sequenced using the Illumina platform. Sequence reads were mapped to the human genome using a customized bioinformatics pipeline. Reads were annotated by BLAT (*Kent, 2002*) to a unified database containing entries for human small RNAs from miRBase (*Griffiths-Jones, 2004*), NONCODE (*Liu et al., 2005*), tRNAs in The RNA Modification Database (*Limbach et al., 1994*), and rRNA entries in the Entrez Nucleotide Database (*Schuler et al., 1996*). Our previous experience performing comparative

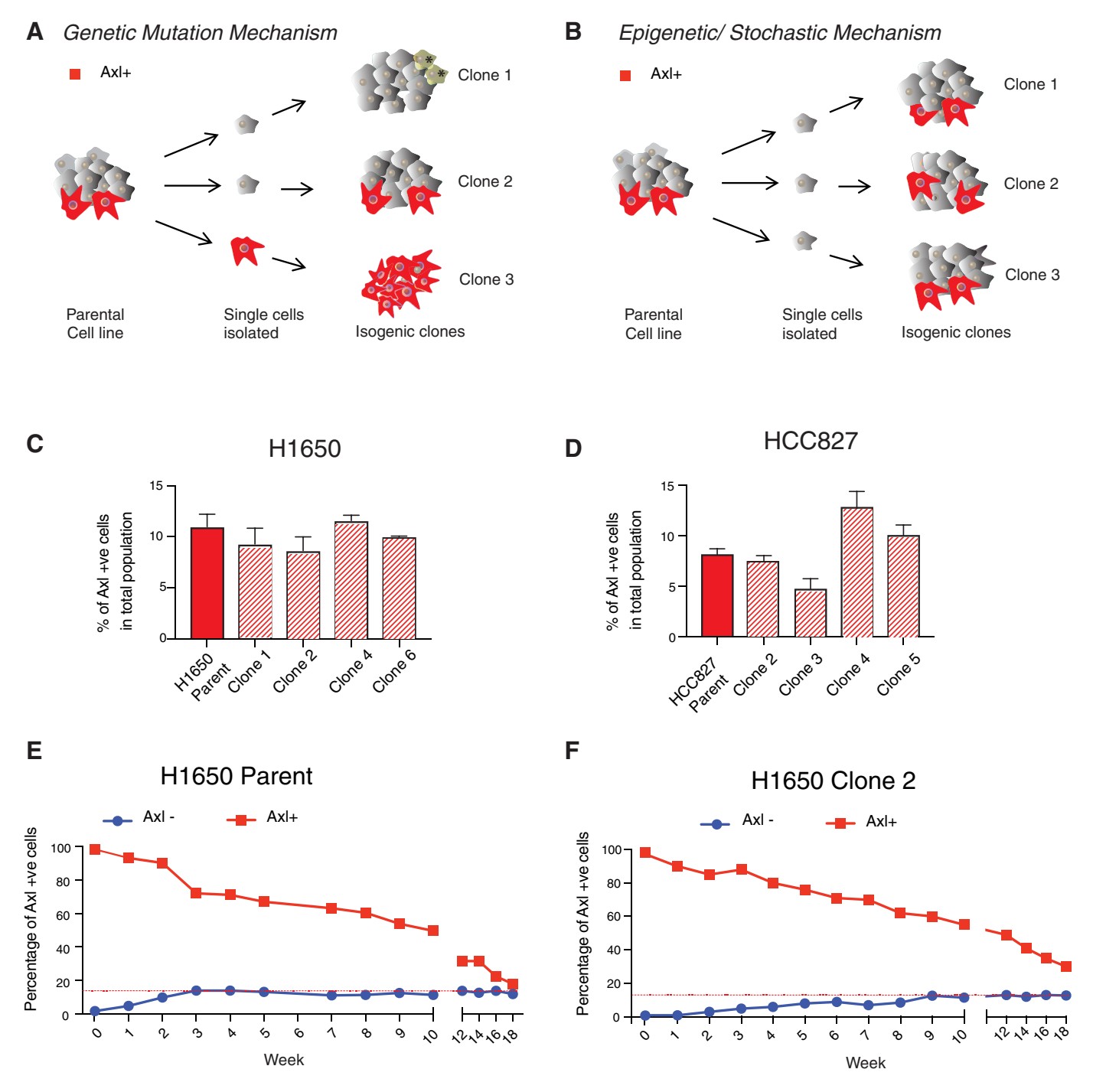

**Figure 3.** AXL-positive cells are generated stochastically. (A) If AXL-positive cells (red) were generated as a consequence of genetic mutations, single-cell-derived clones will have different percentages of AXL-positive cells. (B) On the other hand, if AXL-positive cells were generated stochastically, then an equal percentage of AXL-positive cells will be present in both parental and single cell-derived isogenic clones. (C, D) The charts represent the percentage of AXL-positive cells in the parental and single cell-derived clonal populations in H1650 and HCC827 respectively. Cells were stained for surface expression of AXL, followed by flow cytometry analysis. Each bar represents mean ± SD of three technical replicates from two independent experiments. 20,000 cells were analyzed by FACS for each replicate of each sample. (E) AXL-negative (blue) and AXL-positive (red) cells were sorted from the H1650 cell line and were grown for 18 weeks. The percentage of AXL-positive cells emerging in each population was measured weekly and represented as dots in the chart. The red dotted line represents the percentage of AXL-positive cells that were present in the total H1650 parent cell line. (F) AXL-negative (blue) and AXL-positive (red) cells were sorted from a single-cell-derived clonal cell line from H1650 (H1650 Clone 2) and were

*Figure 3 continued on next page*

*Figure 3 continued*

grown for 18 weeks. The percentage of AXL-positive cells emerging in each population was measured weekly and represented as dots in the chart. The red dotted line represents the percentage of AXL-positive cells that were present in the total H1650 clone 2 cell line.

The online version of this article includes the following figure supplement(s) for figure 3:

**Figure supplement 1.** Parental cell lines and single cell-derived clonal cell lines (H1650 and HCC827) are highly similar from a molecular standpoint.

analysis informed our decision to use an arbitrary cut-off of a minimum of 1000 reads and >2- fold differential expression. Using these criteria, we identified 20 miRNAs that were upregulated and 19 miRNAs that were downregulated in the AXL-positive H1650-M3 cells compared to the AXL-negative H1650 cells (*Figure 4A,B*). Differential miRNA expression levels were independently validated by quantitative stem-loop RT-PCR (qRT-PCR) in the AXL-negative (H1650) and AXL-positive (H1650-M3) cell lines (*Figure 4C*). Apart from let7c, the differential miRNA expression patterns of all miRNAs identified by our deep sequencing analysis were confirmed (*Figure 4C*).

Although none of the identified miRNAs were predicted to target AXL, we were intrigued by the differential expression of miR-335 we observed in AXL-positive cells compared to AXL-negative cells. It has been reported that miR-335 suppresses a mesenchymal-like state and metastatic dissemination by targeting a diverse set of genes regulating cell migration, extracellular matrix remodeling, cell self-renewal, and epigenetic reprogramming (*Tavazoie et al., 2008*; *Figure 1D*). Among them, of particular interest was the regulation of the TGF-β axis by miR-335. In fact, TGF-β is a well-known regulator of AXL and AXL activity (*Lynch et al., 2012*). Furthermore, the TGF-β axis has also been shown to suppress the expression of multiple miRNAs that we found to be downregulated in AXL-positive cells (*Gregory et al., 2011*; *Yang et al., 2012*; *Kato et al., 2013*). Altogether these findings let us to hypothesize that differentially expressed miRNAs in AXL-positive cells could be part of a hierarchically organized miRNA cluster primed by miR-335 and that the regulation of miR-335 could play a major role in the ontogeny of AXL-positive cells.

As a first step to test this possibility, we determined how general was the decrease in miR-335 expression we observed in AXL-positive cells. To this end, we examined the expression of miR-335 in (1) Erlotinib-resistant H1650-M3 and PC14 cells (*Figure 4E*), (2) FACS-sorted AXL-positive and AXL-negative cells from H1650 and HCC827 cell lines (*Figure 4F*), as well as (3) FACS-sorted cells from four human primary NSCLC tumors (*Figure 4G*). In all these cases, when we measured the expression of miR-335 by qRT-PCR, we consistently found that miR-335 levels were decreased in all AXL-positive cells (*Figure 4E–G*).

To verify that miR-335 was active in AXL-positive cells, next, we compared the expression levels of known and predicted miR-335 targets. RT-PCR analysis showed that the miR-335 targets SOX4, TNC, COL1A1, PTPRN2, MERTK, PLCB1, LAMB2, FGF2, JAG1, BMI1, SMARCA2, and MAX were expressed at higher levels in AXL-positive cells; miR-335 low cells (H1650-M3) compared to AXL-negative cells; and miR-335 high cells (H1650) (*Figure 4D*).

We previously have shown that AXL-positive cells have increased activation of the TGF-beta pathway (see *Figure 2*). To determine whether miR-335 was sufficient to regulate the activity of the TGF-β pathway, we inactivated miR-335 by transfecting AXL-negative cells with three independent Antagomirs and assessed the expression of TGF-β 1 and 2 and some of their downstream targets (e.g., Vim, Ecadh, Snail) by RT-PCR. We found that the Antagomirs treatment decreased the expression of miR-335 (*Figure 4—figure supplement 1A*) and of its targets (*Figure 4—figure supplement 1B*) as well as of TGF-β 1/2 (*Figure 4—figure supplement 1C*) and of the TGF-β target genes VIM, Ecadh, SNAI, SLUG, etc. compared to control (*Figure 5D*).

As reported in the literature, we also observed the majority of miRNAs we observed to be differentially expressed in AXL-positive and AXL-negative cells to be regulated by TGF-β 1/ two except for MiR-335 (*Figure 4H,I*). Consistent with TGF-beta being regulated by miR-335, we also found that inactivation of miR-335 was sufficient to reduce the expression of these miRNAs (miR-20a, miR-34a, miR-200c, etc.) but to increase the expression of miR-143 and miR-195, which were expressed at higher levels in AXL-positive cells when compared to AXL-negative cells (*Figure 4J*).

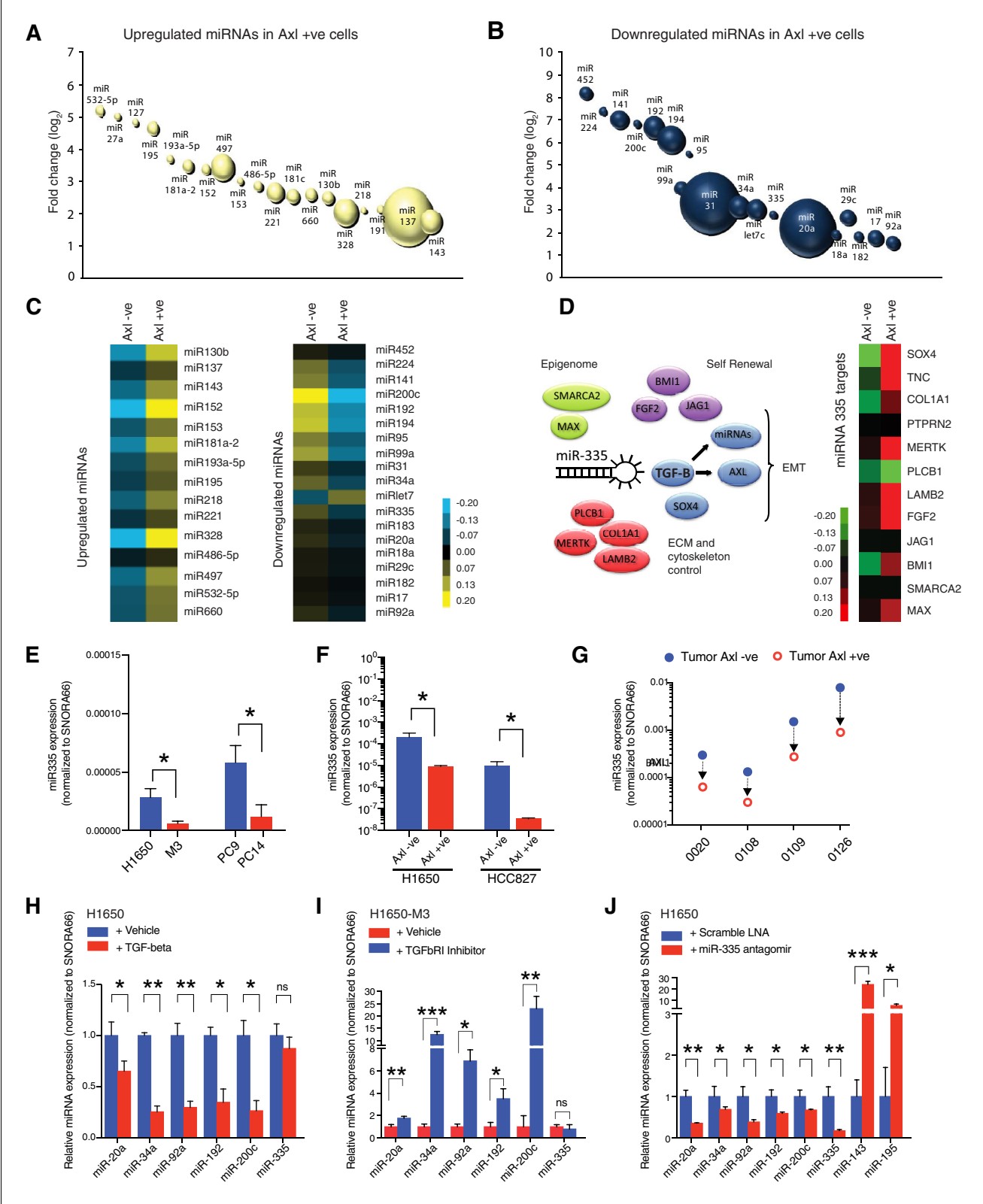

**Figure 4.** miRNA profiling reveals a distinct signature that characterizes the AXL-positive cell state. (A, B) The bubble charts show miRNAs that were >2-fold upregulated or >2-fold downregulated in AXL-positive (H1650-M3) cells relative to the parental AXL-negative (H1650) cells. Small RNA libraries were generated from each cell line and sequenced using an Illumina platform. The size of the bubble represents the abundance of the miRNA. (C) Heat map depicts patterns of miRNA expression in AXL-negative (H1650) and AXL-positive (H1650-M3) cells, validated by quantitative stem-loop RT-PCR.

*Figure 4 continued on next page*

*Figure 4 continued*

Columns indicate relative expression changes compared to U6 snRNA. Each square represents the average of three independent measurements. p≤0.0001, unpaired t-test. (D) miR-335 targets are increasingly expressed in AXL-positive cells. The heat map on the left shows changes in mRNA expression of miR-335 targets in AXL-negative (H1650) and AXL-positive (H1650-M3) cells. Each column represents changes in mRNAs expression relative to Actin. Each square represents the average of three independent measurements. p≤0.0001, unpaired t-test. (E) The chart represents the expression of miR-335 normalized to SNORA66 in the indicated cell lines. miRNA expression was quantified by ExiLENT SYBR-green-based RT-qPCR. Each bar represents mean ± SD of three replicates from two independent experiments (*p<0.05, unpaired t-test). (F) The chart represents expression of miR-335 normalized to SNORA66 in AXL-negative (blue) and AXL-positive (red) cells sorted from H1650 and HCC827 cell lines. miRNA expression was quantified by ExiLENT SYBR-green-based RT-qPCR. Each bar represents mean ± SD of three replicates from two independent experiments (*p<0.05, unpaired t-test). (G) The chart represents expression of miR-335 normalized to SNORA66 in AXL-negative (blue) and AXL-positive (red) cells sorted from four human primary NSCLC tumors. miRNA expression was quantified by ExiLENT SYBR-green-based RT-qPCR. Each dot represents mean ± SD of three replicates. (H) The chart represents expression of the indicated miRNAs normalized to SNORA66 in AXL-negative H1650 cells treated with Vehicle (blue) or TGF-beta (red). The data are presented as relative to vehicle-treated control. miRNA expression was quantified by ExiLENT SYBR-green-based RT-qPCR. Each bar represents mean ± SD of three replicates from two independent experiments (*p< 0.05, **<0.005, paired t-test). ns = non-significant. (I) The chart represents expression of the indicated miRNAs normalized to SNORA66 in AXL-positive H1650-M3 cells treated with vehicle (red) or TGFbRI inhibitor LY2157299, Selleckchem (red). The data are presented as relative to vehicle-treated control. miRNA expression was quantified by ExiLENT SYBR-green-based RT-qPCR. Each bar represents mean ± SD of three replicates from two independent experiments (*p<0.05, **<0.005, ***<0.005 paired t-test). ns = non-significant. (H) The chart represents expression of the indicated miRNAs normalized to SNORA66 in AXL-negative H1650 cells treated with Scramble LNA (blue) or miR-335 antagomir (red). The data are presented as relative to scramble-treated control. miRNA expression was quantified by ExiLENT SYBR-green-based RT-qPCR. Each bar represents mean ± SD of three replicates from two independent experiments (*p<0.05, **<0.005, paired t-test).

The online version of this article includes the following source data and figure supplement(s) for figure 4:

**Source data 1.** miRNA sequence read-counts and log fold change.
**Figure supplement 1.** Inhibiting miR-335 expression results in re-expression of miR-335 targets.

## miR-335 regulates the ontogeny of AXL-positive cells

Our data indicated that miR-335 regulates the expression of key molecular determinants of the AXL-positive state. To test whether miR-335 could regulate the ontogeny of AXL-positive cells, we decreased the expression of miR-335 using Antagomir treatment in multiple AXL-negative cells and analyzed the morphology, the expression of signature genes, as well as their resistance to EFGR Tki.

We observed that treatment of AXL-negative cell lines (H1650 and PC9) with a miR-335 Antagomir resulted in a reduction of miR-335 expression (*Figure 5A*) and an increased expression of AXL-positive cells (*Figure 5B*, *Figure 5—figure supplement 1A*). This was accompanied by epithelial-to-mesenchymal transition manifested by loss of the classic cobblestone appearance of epithelial cells (*Figure 5C*) and changes in EMT molecular markers (*Figure 5D*).

In a standard drug sensitivity assay, we observed that treatment with miR-335 Antagomir also increased the resistance of cells to Erlotinib treatment to levels similar to what we observed when we tested the AXL-positive cell lines we derived by Erlotinib selection (*Figure 5E,F*).

To provide additional proof that inhibition of miR-335 was sufficient for the generation of AXL-positive cells, we utilized CRISPR-CAS9 gene editing as an orthogonal approach. Also in this case and consistent with our previous results, genetic inactivation of miR-335 resulted in the acquisition of phenotypic and molecular characteristics of AXL-positive cells (*Figure 5—figure supplement 2A–C*).

Interestingly, when we express miR-335 mimic oligonucleotides in the AXL+ cells H1650-M3 and PC14, we observed a dramatic decrease in cell viability (*Figure 5G–H*). To exclude this was due to a non-specific effect of miRNA mimic oligonucleotide transfection, we repeated the same experiment by transfecting an unrelated miRNA (*Figure 5G–H*, *Figure 5—figure supplement 3A–B*). In this case, there was no difference in the cell viability compared to control (transfection agent). Hence, the transition of cells into AXL+ cell state induced the rewiring of cell signalings to which cells become 'addicted'.

Altogether, these observations indicate that miR-335 serves as a critical regulator of the interconversion of AXL-negative and AXL-positive cell states beyond its well-studied role in the regulation of metastasis (*Tavazoie et al., 2008*).

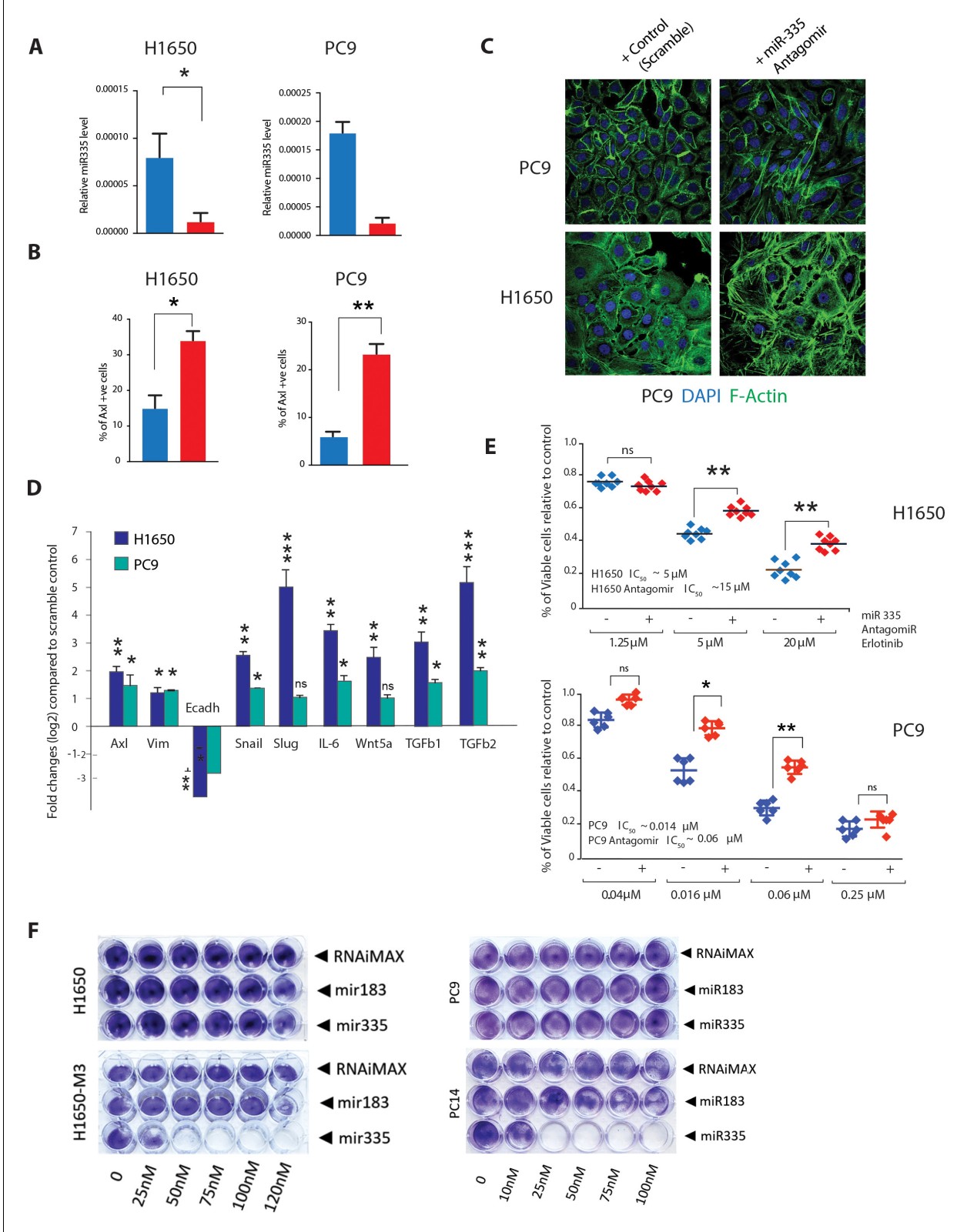

**Figure 5.** miR-335 regulates AXL-positive cell-state transition. (**A**) The chart on left represents the knockdown efficiency of miR-335 antagomir in H1650 and PC9 cells. mRNA expression was quantified by SYBR-green-based RT-qPCR and normalized to GAPDH. Each bar represents mean ± SD of three replicates from two independent experiments (*p<0.05, unpaired t-test). (**B**) The chart represents the percentage of AXL-positive cells present in H1650 and PC9 cells upon treatment with miR-335 antagomir relative to scramble-treated control. Cells stained with antibody against N-terminal of AXL

*Figure 5 continued on next page*

*Figure 5 continued*

were used for FACS analysis. Each bar represents mean ± SD of three technical replicates. 20,000 cells were analyzed by FACS for each replicate of each sample (*p<0.05, unpaired t-test). See *Figure 5—figure supplement 1A* for representative FACS profiles. (**C**) Representative images of H1650 and PC9 cells stained with Phalloidin (green) and DAPI (blue) as counter-stain. Cells were treated for 5 days with control antagomir or miR-335 antagomir. (**D**) Fold change in genes that characterize the AXL-positive cell state upon inhibition of miR-335 in AXL-negative cell lines H1650 and PC9. mRNA expression was quantified by SYBR-green-based RT-qPCR and normalized to actin. Each bar represents mean ± SD of three replicates from two independent experiments (*p<0.05, **<0.005, ***<0.0005 unpaired t-test). (**E**) The charts represent the number of viable cells in H1650 and (**F**) PC9 cells upon transfection with miR-335 antagomir and treatment with indicated doses of Erlotinib. Values are normalized relative to vehicle-treated cell (control). Cells were grown for 120 hr in the presence of the drug; the number of cells was estimated upon staining with the crystal violet, de-staining in 100 µl of 10% acetic acid, and reading absorbance at 590 nm. Diamonds and black bars represent single-point measurements and the mean, respectively (n = 8) (**p<0.005, unpaired t-test). ns = non-significant. (**F**) Representative pictures of a cell viability assay by crystal violet staining. Cells (H1650, H1650-M3, PC9, and PC14) were plated in 24-well plates and transfected with miR-183 and miR-335 mimic oligonucleotides or with the transfecting agent RNAiMAX alone as indicated. The cells were then stained with crystal violets 96 hr after transfection. Quantification of the experiment is provided in *Figure 5—figure supplement 3*.

The online version of this article includes the following figure supplement(s) for figure 5:

**Figure supplement 1.** Inhibiting miR-335 expression results in molecular and phenotypic changes characteristic of the AXL-positive cell state.

**Figure supplement 2.** CRISPR-CAS9 mediated gene editing to reduce miR-335 expression results in molecular and phenotypic changes characteristic of the AXL-positive cell state.

**Figure supplement 3.** The charts represent the quantification of a cell viability assay by crystal violet illustrated in *Figure 5F*.

## Methylation of *MEST* isoform 2 promoter modulates miR-335 expression in AXL-positive cells

The miR-335 encoding sequence resides in the second intron of the *mesoderm-specific transcript homolog* (*MEST*)/*paternally expressed 1* (*PEG1*) gene located on chromosome 7q32. In humans, two distinct CpG islands have been identified in the promoters of *MEST* (*Figure 6A*; *Png et al., 2011*; *Dohi et al., 2013*). To investigate the possible epigenetic regulation of miR-335, we analyzed levels of *MEST* CpG island 1 and 2 methylation by bisulfite sequencing, methylation-specific RT-PCR, as well as qRT-PCR in AXL-positive H1650-M3 and AXL-negative H1650 cells (*Figure 6A,B*). We found that although no significant differences were observed in the methylation of CpG island 2, CpG island 1 was differentially methylated in the AXL-positive H1650-M3 cells and associated with higher expression of MEST isoform 1 and decreased expression of miR-335 (*Figure 6B*, *Figure 6—figure supplement 1A*).

We extended these analyses to include AXL FACS-sorted cell lines (H1650 and PC9) and human NSCLC tumor-derived cells. Again, we found that all AXL-positive cells displayed increased methylation of CpG island 1 relative to AXL-negative cells (*Figure 6C,D*, *Figure 6—figure supplement 1B*).

To establish the functional relevance of the hypermethylation of the *MEST* isoform 2 promoter, we treated AXL-negative H1650 cells and AXL-positive H1650-M3 cells with the DNA methylation inhibitor 5-aza-2'-deoxycytidine (5-Aza-dC). Consistent with the observation that re-expression of mirR-335 in AXL+ cells resulted in cell death, long-term treatment with 5-Aza-dC in AXL+ cells H1650-M3 and PC14 revealed an increased sensitivity of these cells to the drug treatment when compared to the AXL− cells H1650 and PC9 (*Figure 6—figure supplement 1C,D*). To determine whether this could be due to changes in the methylation of MEST promoter, we examined the methylation status of MEST CpG island 1 upon short treatment (36 hr) with 5-Aza-dC in H1650 and H1650-M3 cells. We observed a dose-dependent change in the methylation of CpG island 1 and, consistent with the role of CpG island hypermethylation in gene silencing, increased expression of miR-335 in AXL-positive cells (*Figure 6E,F*). Importantly, no differences were observed in the AXL-negative H1650 cells upon treatment with 5-Aza-dC compared to the control. Following the proposed role of miR-335 in the regulation of AXL, we also observed a decrease in AXL mRNA expression (*Figure 6—figure supplement 1C*). These changes were most likely due to increased miR-335 levels as inhibition of miR-335 by Antagomir treatment impeded the observed decrease in the number of AXL-positive cells in H1650-M3 cells (AXL-positive) treated by 5-Aza-dC (*Figure 6G*).

To further characterize a possible role of miR-335/MEST DNA methylation in Erlotinib resistance, we performed a drug sensitivity assay in which we combined 5-Aza-dC and Erlotinib treatment. Consistent with 5-Aza-dC decreasing MEST promoter methylation and the viability of AXL-positive cells (*Figure 6E–G*), we observed a decrease in AXL-positive cells in H1650 cells upon 5-Aza-dC

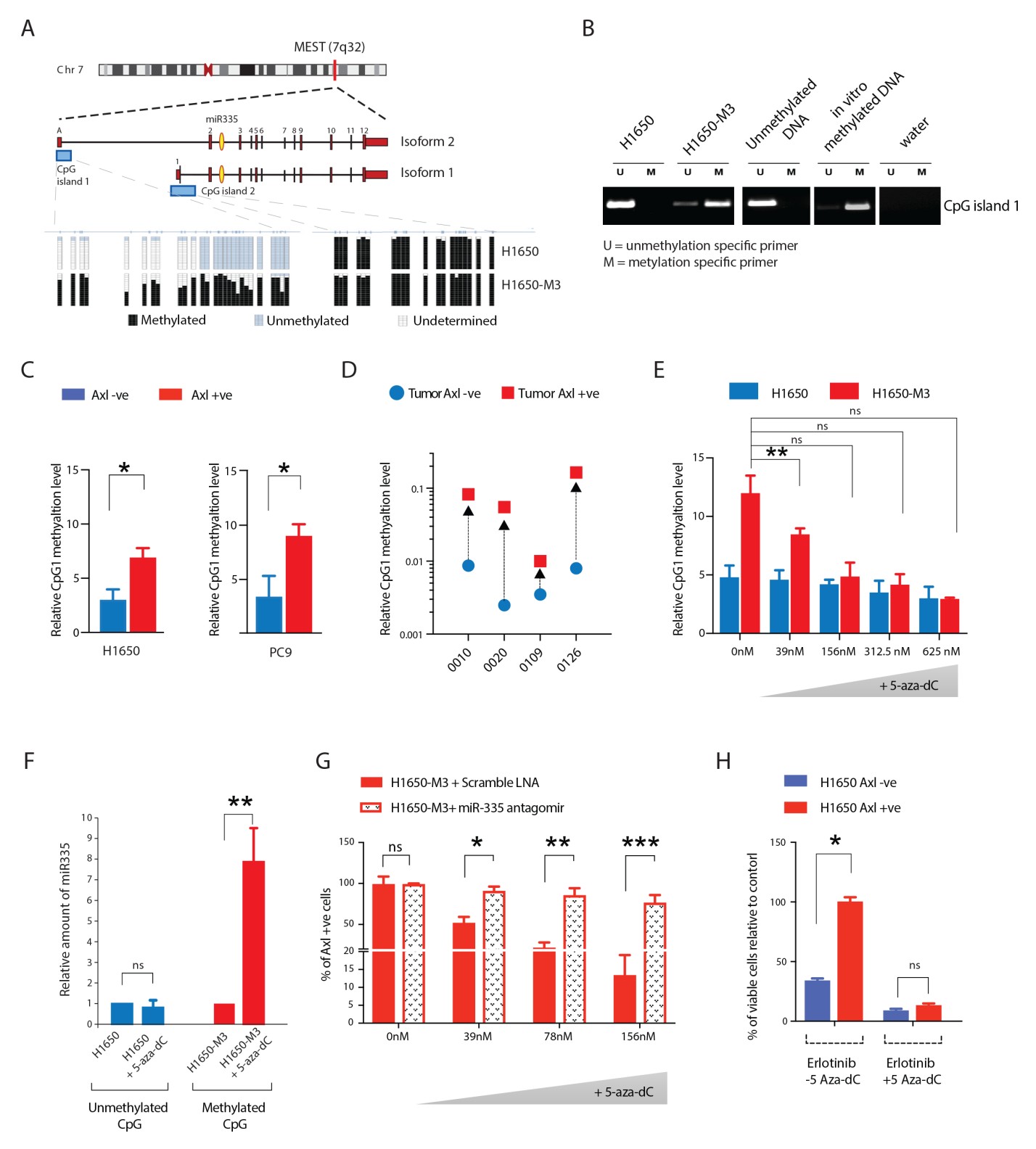

**Figure 6.** Methylation of MEST isoform 2 promoter modulates miR-335 expression in AXL-positive cells. (**A**) Schematic of MEST locus organization and the methylation analysis by bisulfite sequencing of MEST CpG island 1 and 2 in AXL-negative (H1650) and AXL-positive (H1650-M3) cells lines (lower panel). Each box indicates a CpG dinucleotide, and each line of boxes represents the analysis of a single cell. The color of each box indicates the methylation of each CpG island. (**B**) Semi-quantitative RT-PCR analysis of methylation status of CpG island 1 in H1650 and H1650-M3. U: Unmethylation-

*Figure 6 continued on next page*

*Figure 6 continued*

specific primer; M: Methylation-specific primer. See *Figure 6—figure supplement 1A* for the relative amount of methylation of MEST CpG island 1 in the Erlotinib-naïve and -resistant cell lines quantified via methylation-specific SYBR-green-based RT-qPCR (MSP). (C) The chart represents the relative amount of methylation of MEST CpG island 1 in AXL-negative (blue) and AXL-positive (red) cells sorted from H1650 and PC9 cell lines. The amount of methylation of DNA was quantified via methylation-specific SYBR-green-based RT-qPCR (MSP). Each bar represents mean ± SD of three replicates from two independent experiments (*$p<0.05$, unpaired t-test). (D) The chart represents the relative amount of methylation of MEST CpG island 1 in AXL-negative (blue) and AXL-positive (red) cells sorted from four human primary NSCLC tumors. The amount of methylation of DNA was quantified via methylation-specific SYBR-green-based RT-qPCR (MSP). Each dot represents mean ± SD of three replicates. (E–H) Treatment of cells with 5-Aza-dC for is sufficient to reduce the levels of CpG island 1 methylation, increase the expression of miR-335, decrease AXL-positive cells, and reduce Erlotinib resistance, respectively. (E) The chart represents the relative amount of methylation of MEST CpG island 1 in H1650 and H1650-M3 cells treated with the indicated amount of 5-Aza-dC for 36 hr. The amount of methylation of DNA was quantified via methylation-specific SYBR-green-based RT-qPCR (MSP). Each bar represents mean ± SD of 3 replicates from two independent experiments (*$p<0.05$, unpaired t-test). ns = non-significant. (F) The chart represents the amount of miR-335 (relative to SNORA66) in H1650 and H1650-M3 cells upon treatment with 5-Aza-dC for 36 hr and normalized to vehicle-treated control. miRNA expression was quantified by ExiLENT SYBR-green-based RT-qPCR. Each bar represents mean ± SD of three replicates from two independent experiments (*$p<0.05$, unpaired t-test). ns = non-significant. (G) The chart represents the percentage of AXL-positive cells in H1650-M3 upon treatment with 5-Aza-dC for 96 hr, in the presence of scramble LNA (red solid bar) or miR-335 antagomir (dotted bar). The data is presented relative to H1650-M3 cells treated with scramble LNA at 0 nM 5-Aza-dC. Cells stained with the antibody against N-terminal of AXL were used for FACS analysis. Each bar represents mean ± SD of three technical replicates. 20,000 cells were analyzed by FACS for each replicate of each sample (*$p<0.05$, **$p<0.005$, ***$p<0.05$ unpaired t-test). ns = non-significant. See *Figure 6—figure supplement 1C* for the representative mRNA expression. (H) The chart represents the relative number of Erlotinib surviving cells in theabsence or in the presence of 5-Aza-dC. Each bar represents mean ± SD of three replicates from two independent experiments (*$p<0.05$, unpaired t-test). ns = non-significant.

The online version of this article includes the following figure supplement(s) for figure 6:

**Figure supplement 1.** Methylation of MEST isoform 2 promoter moduates miR-335 expression in AXL-positive cells.

treatment (*Figure 6—figure supplement 1D*) and overall increased sensitivity of H1650 cells to Erlotinib (*Figure 6H*).

Based on these observations, we concluded that differential miR-335 promoter methylation is responsible for the decreased expression of miR-335 observed in AXL-positive cells and that the transition between the AXL-positive and AXL-negative cell states as well as their differential resistance to EGFR TKi is regulated epigenetically.

## Discussion

Drug resistance continues to be a major hurdle that oncologists face in treating cancer patients. Although the genetic diversity of tumors has been proposed to drive the acquisition of drug resistance; emerging data indicate that also non-genetic determinants could be equally significant (*Brock et al., 2009*). These include the interaction of a tumor with its micro-environment as well as the occurrence of cell-intrinsic molecular mechanisms such as epigenetic changes (*Brock et al., 2009*; *Muranen et al., 2012*).

In the case of lung tumors driven by oncogenic-EFGR mutations, it has been observed that approximately 15% of tumors that become resistant to EGFR TKi are characterized by mesenchymal-like features and higher expression of AXL. In these tumors inhibition of AXL restore the sensitivity to EGFR TKi (*Zhang et al., 2012*; *Taniguchi et al., 2019*). In contrast to other mechanisms of resistance to EGFR TKi, in these tumor cells, AXL was neither mutated, amplified, nor its expression driven by EGFR TKi treatment as in the case of the persistent cells originally described by *Sharma et al., 2010*.

Here we described a novel molecular mechanism driving the ontogeny of AXL-positive EGFR TKi-resistant cells based on the stochastic fluctuation of cancer cells between an AXL-negative state characterized by epithelial-like features and an AXL-positive state in which cells are mesenchymal and have an increased resistance to EGFR TKi (*Figure 7*). The switch between these two cell states is restricted by miR-335 as all AXL-positive cells we examined were characterized by a decreased expression of miR-335 and that inactivation of miR-335 decreased the number of AXL-positive cells and reverted AXL-positive cells into AXL-negative cells.

Although miR-335 restricts the transition of AXL-positive into AXL-negative cells, AXL mRNA does not contain a miR-335 seeding sequence, which means it is unlikely to be a direct target of miR-335. Yet, miR-335 has been previously shown to regulate the expression of a multitude of

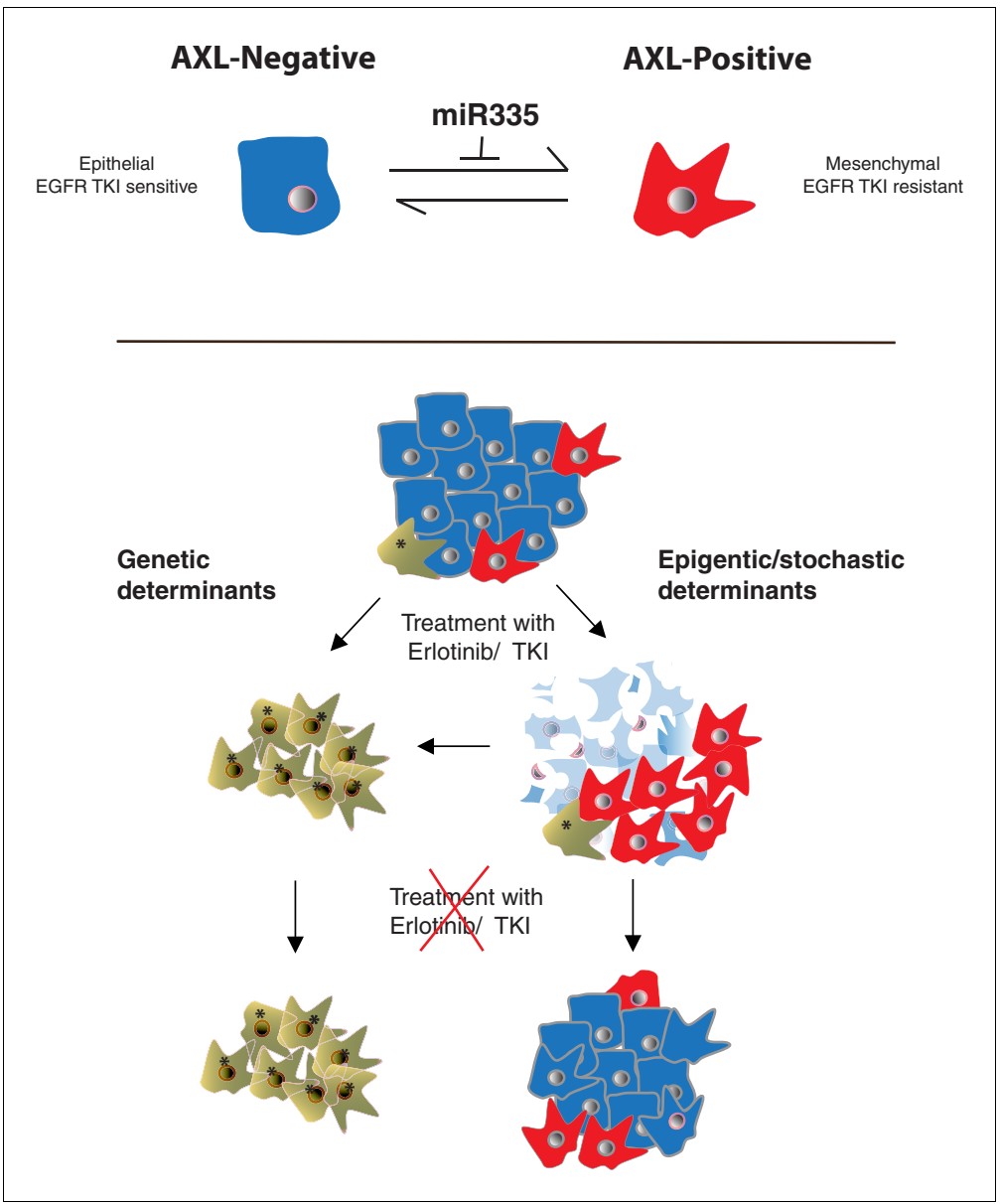

**Figure 7.** Cancer cells can transit between an epithelial state characterized by low expression of AXL and a mesenchymal-like state with high AXL expression. AXL-positive cells have increased resistance to EGFR TKi compared to AXL-negative cells. The transition between these two states is restricted by miR-335 whose expression is regulated epigenetically through promoter methylation. The existence of this innate stochastic/epigenetic mechanism has important therapeutic implications. Upon treatment with EGFR TKi, AXL-positive cells can survive but differently from cells that have acquired resistance through genetic mutations with time they can revert to an epithelial EGFR TKi-sensitive state.

signaling pathways including components of the TGF-β axis. The regulation of the TGF-β axis by miR-335 is of particular interest because AXL is a known downstream target of TGF-β, hence suggesting a possible molecular linking AXL and miR-335 (*Bauer et al., 2012*).

The inhibition of the TGF-β signaling pathway by miR-335 is particularly interesting also because exposure to TGF-β regulate the expression of several of the miRNAs associated with the AXL-positive state, such as miR-20a, miR-34a, miR-200c, etc. Among them of particular significance were the mir-200 family members, as they not only induce EMT but also resistance to Erlotinib (*Brabletz and Brabletz, 2010*).

Lastly, miR-335 is among approximately 50 miRNAs that are regulated epigenetically by DNA methylation of CpG islands within promoter regions (*Lujambio and Esteller, 2007*). The epigenetic regulation of miR335 is particularly interesting in light of the observation that AXL-positive and AXL-negative cell populations are highly dynamic. AXL FACS-sorted cells can interconvert until the same two cell state distribution observed in the parental cell population is reached (*Figure 3E*). Notably, we observed the time required for AXL-positive and AXL-negative cell populations to reach equilibrium is different. While within a few weeks the AXL-negative cells generated a population of cells with the same distribution of AXL-positive and AXL-negative cells as was observed in the parental cell line; it took the AXL-positive cells a couple of months to reach this equilibrium. At this time, we do not have a clear molecular mechanism explaining these differences, yet it is tentative to hypothesize that – `given` the stochastic regulation of miR335 – because the AXL-positive cells grow slower than AXL-negative cells, it will take a longer time for the former to switch state. Alternatively, it is tempting to postulate that the interconversion between the two cell states could be regulated by enzymatic activities occurring at different rates. This hypothesis is informed by the possibility that the de-methylation rate of the CpG1 on the miR-335 promoter could occur less efficiently than its methylation.

One important feature of AXL-positive cells is their intrinsic resistance to EGFR TKi. This implies that because AXL-positive cells could revert back to an AXL-negative state, the drug resistance observed in AXL-positive tumors, although heritable, is not a stable trait in the population. Consequently, as shown in the schematic in *Figure 7*, AXL-negative cells could hypothetically emerge over time following drug removal. This phenomenon is in principle similar to the observation that certain Erlotinib-resistant tumors expressing neuroendocrine markers can revert to an epithelial-like state over time upon interruption of the drug treatment (*Niederst et al., 2015*).

Nevertheless, the AXL-positive cells that survived to the drug treatment can accumulate novel genetic mutations that can alter the innate equilibrium between AXL-positive and AXL-negative and, consequently, produce tumors that are stable AXL-positive state (*Figure 7*). This explanation seems to be true for certain clonal populations, including the H1650-M3 and PC14 cells that were selected to grow and expand in the presence of high Erlotinib concentrations. In fact, these cell lines maintain features associated with the AXL-positive cell state even in the absence of drug treatment. Interestingly, we have recently observed that TGF-beta by repressing DNA repair could spur the accumulation of mutations and accelerate the clonal evolution of tumors (*Pal et al., 2017*).

Interestingly, the reactivation of miR-335 resulted in a dramatic decrease in the viability of AXL+ cells, suggesting that the cells transiting into AXL+ cells are addicted to specific singling, regulated by miR-335. Altogether these findings have important clinical implications. They predict that treatments based only on targeting the epithelial, AXL-negative cells, such as in the case of Erlotinib and Osimertinib treatments, will be insufficient and poised to fail. Yet, combinatorial treatments targeting both cell states could increase the sensitivity of drug treatment and slow or prevent the acquisition of tumor resistance. This could be the case of co-treatment of tumors with EGFR TKi and AXL inhibitors or EGFR TKi and 5-Aza-dC. Of note, our in vitro studies indicated that the concentration of 5-Aza-dC to which AXL-positive cells are sensitive are well within the 5-Aza-dC blood concentrations observed in clinical trials for the treatment of different cancer types (*Karahoca and Momparler, 2013*).

## Materials and methods

### Cell culture

H1650, HCC4006, HCC827, A549, H358, and H2228 cell lines were purchased from American Type Culture Collection. All cell lines were authenticated through short tandem repeat (STR) profiling and regularly tested for mycoplasma. The PC14 cell line was obtained by Dr. Kazuto Nisho (National Cancer Center Hospital, Tokyo). H1650-M3 cells were generated by culturing the H1650 cell line in the presence of a constant high concentration of Erlotinib as previously described (Yao et al.). All cell lines were maintained in RPMI GlutaMAX (Invitrogen) containing 5% fetal bovine serum. The culture medium was supplemented with 100 units/ml of penicillin and 100 ug/ml of Streptomycin (Invitrogen). All cell lines were cultured at 37°C and 5% $CO_2$.

## Generation of isogenic clones

H1650 cells were serially diluted in 96 wells such that one well contains one cell. They were then grown for 2 months before the experiments.

## TGF-β treatment

Cells were treated with rhTGFβ1 and rhTGFβ2 (R and D Systems, Minneapolis, MN) 1 ng/ml each in complete media, for 72 hr. Following treatment, the cells were harvested for RNA preparation and qRT-PCR, for immunoblotting, or for cell cycle analysis.

## TGF-β inhibitor treatment

For treatment with LY2157299 (20 μM) (TGFBR1 kinase inhibitor, Selleckchem, Houston, TX), 300,000 H1650-M3 cells were plated in a 6 cm$^2$ plate. The inhibitor was added the next day, and the mixture was incubated for 3–5 days for LY2157299. The cells were lysed with TRIzol and processed for RNA preparation.

## Drug treatment

To determine IC$_{50}$ values for various drugs (Erlotinib and BMS-777607), the cells were plated in 96-well plates at 1000 cells/well. The next day, individual drugs were added to the wells at the indicated concentrations and incubated for 5 days. The plates were then washed once with PBS, fixed with 3.7% formaldehyde, and stained with Crystal violet. Each stained well was de-stained in 50–100 μl of 10% acetic acid, and the absorbance was read in a spectrophotometer at 590 nm.

## Long-term drug treatment to generate persisters

H1650-M3 cells were generated according to the protocol previously described by *Yao et al., 2010*.

## RNAi transfection, RNA extraction, and quantitative real-time PCR

RNAi transfection was performed using Lipofectamine 2000 (Invitrogen) as per the manufacturer's protocol. Unless otherwise indicated, total RNA was collected 72 hr after transfection. Total RNA was extracted using Trizol (Life technologies). Removal of contaminating genomic DNA was performed by incubation with RQ1 RNase-Free DNase (Promega) for 30 min. One thousand nanogram total RNA was reverse transcribed using ImProm-II reverse transcriptase and Oligo-dT primers. Quantitative PCR was carried out using Power SYBR Green PCR master mix on a 7900HT Fast Real-Time System (Applied Biosystems) or QuantStudio-6 Real-Time System (Applied Biosystems). Power SYBR Green Cells-to-Ct kit was used to perform quantitative PCR on 10,000 cells sorted from tumors. Analyses were done in triplicate, and Actin or GAPDH was used as a reference gene. A complete list of primer sequences is supplied in *Supplementary file 1*.

## Lentiviral infection

Packaging cells HEK293T in passage six were seeded evenly at density 1–2 × 10$^6$ cells per 10 cm plate and incubated in DMEM supplemented with 10% FBS in the incubator, at 5% CO$_2$ and 37°C for 20 hr. When they reached 70% confluency, the media was changed 1 hr before the transfection to a final volume of 10 ml. The mixture of two tubes, each with 500 μl of warm OptiMEM media, was prepared. In one tube, 25 μl of lipofectamine 2000 transfection reagent (Thermo Fisher cat # 11668019) was added and incubated for 5 min. To the second tube, a mix of plasmids was added – 10 μg pLenti CMV Puro DEST (w118-1) eGFP (Addgene Plasmid #107505), 6 μg of psPAX2, and 3 μg of pMD2.G plasmid. Tube one was mixed into tube two to form a transfection mixture and incubated for 20 min at room temperature. The mixture was dropwise to HEK293T cells and plates were incubated overnight. After 18 hr, the media was changed to 10 ml of fresh DMEM with 10% FBS and plates were incubated for the next 48 hr before the collection and filtering of the supernatant through a 0.45 μm filter was performed. Polybrene reagent (8 μg/ml) was added to the virus before infecting PC9 and H1650 cells for 24 hr. The flow cytometer (Guava easyCyte Flow Cytometer) was used to determine the percentage of GFP expression.

## miRNA analysis

Total RNA was extracted using miRCURYRNA isolation kit – cell and plant (Exiqon) according to manufacturer's instructions. On-column removal of genomic DNA was performed using RQ1 RNase-Free DNase (Promega). cDNA synthesis was performed using miRCURY LNA Universal RT microRNA PCR, Polyadenylation and cDNA synthesis kit (Exiqon; 203300) and miRCURY LNA Universal RT microRNA PCR SYBR green master mix was used for quantitative real-time PCR analysis. U6 snRNA was used as a reference gene. A complete list of primer sequences is supplied in *Supplementary file 1*.

## Methylation-specific RT-PCR, QPCR, and bisulfite sequencing

One microgram genomic DNA was pretreated with sodium hydroxide for 15 min at 37°C followed by incubation with hydroquinone (Sigma) and sodium metabisulfite (Sigma) for 16 hr at 50°C. The bisulfite modified DNA was subsequently purified using Wizard DNA Clean-up system (Promega). Genomic DNA from in vitro methylated Jurkat cells was (Active Motif) served as positive control and genomic DNA from H1993 cells served as a negative control. RT-PCR analysis was performed using Immolase DNA polymerase (Bioline), and fragments were separated on 2% agarose gels. Takara Episcope MSP kit was used for performing quantitative RT-PCR on a 7900HT Fast Real-Time System (Applied Biosystems). For analysis by bisulfite sequencing, fragments were cloned into the pGEM-T Easy Vector (Promega), and 20 colonies from each sample were sequenced.

## Immunofluorescence

AXL-negative and AXL-positive cells from H1650 and PC9 were FACS-sorted and cultured for 2 days in an 8-well chamber slide system (LAB-TEK, Thermo Fisher Scientific). H1650, H1650-M3, PC9, and PC14 cells were grown on glass coverslips in a 24-well Petri dish. Cells were fixed with 4% paraformaldehyde and permeabilized in 0.1% Triton X-100 in PBS for 10 min. Fixed cells were washed three times in PBS and blocked with 1% BSA in PBS for 1 hr. After washing three times with PBS, the cells were incubated with Alexa Fluor 488 Phalloidin for 30 min at room temperature. DAPI was used for nuclear staining. The stained cells were mounted with a Vectashield mounting medium (Vector Laboratories, Burlingame, CA) and analyzed using a confocal microscope.

## Immunoprecipitation and western blot analysis

Total cell lysates were obtained by lysing cells in modified denaturing buffer (50 mM Tris–Cl pH 7.5, 1 mM EDTA, 1 mM EGTA 1% Triton-X, 0.27 M sucrose, 1% β-mercaptoethanol) with protease inhibitor tablets and phosphatase inhibitors (10 mM NaF, 1 mM PMSF, 1 mM $Na_3VO_4$). Lysate were incubated on ice for 30 min, mixed end-to-end at 4°C for 30 min and then centrifuged at 13,000 g for 30 min to remove debris. One thousand five hundred microgram total protein lysate was 100 µl slurry of pre-cleared with Protein G agarose beads (Promega), followed by overnight incubation with AXL antibody (2 µg Ab/ 100 µl lysate). Immunocomplexes were pulled down by incubating with 100 µl slurry of pre-cleared with Protein G agarose beads. Immunoprecipitation complex and 10% lysate inputs were separated on 8% polyacrylamide gels, transferred to a nitrocellulose membrane, and blotted overnight with antibody against AXL, Phospho-AXL, Phospho-Tyrosine. β-tubulin was used as a loading control.

## Flow cytometry

Cells were dissociated using TrypLE (Invitrogen) and washed with cold PBS containing 5% fetal bovine serum. Resuspended cells were filtered through a 40-micron mesh to generate single-cell suspension and incubated with directly conjugated fluorescent antibodies to the desired antigens for 20 min on ice in the dark and subsequently washed three times with cold PBS pH 7.2. Analysis of AXL-negative and AXL-positive cell populations was performed on the LSRII (BD Biosciences). A total of 20,000 cells were analyzed using the FACSDiva 6.0 software (BD Biosciences). Isolation of AXL-negative and AXL-positive cells were done by fluorescence-activated cell sorting performed on the Aria II (BD Biosciences). For sorting cells from tumor, we stained a single-cell suspension derived from tumors with CD45, CD31, EpCAM, and AXL antibodies. Based on the isotype staining, we gated the CD45$^-$/CD31$^-$/EpCAM$^{mid/high}$ population and then gated the desired AXL-negative and AXL-positive populations from the EpCAM $^{mid/high}$ population.

## Statistical analysis

Data are represented as mean ± SD. Statistical analysis of experimental data was conducted using GraphPad Prism 7.0 software (San Diego, CA). Student's t-test (two-tailed) was used for two-group comparisons. Spearman's rank test was used to measure the correlation between two variables. $p < 0.05$ was considered statistically significant.

## Patient study details

The collection of human lung tissue samples and blood for this study was covered by Northwell Health/Cold Spring Harbor Laboratory IRB #TDP-TAP 1607 (Raffaella Sordella/10/11/16). The samples were acquired from patients already undergoing thoracic procedures (e.g., surgical tumor resection, biopsy) at Huntington Hospital. All study participants provided informed consent for the use of their lung tissue and blood for research purposes. Participants were informed of study aims, the potential risks, and benefits of participation, and that any discoveries facilitated by the analysis of their tissues might be published. The participants were informed that their names would not be associated with their samples in any publication or presentation of research findings.

## Reagents

Recombinant human TGF-β1 and TGF-β2 was purchased from R and D Systems. miR-335 antagomirs were obtained from the following companies: antagomir one from Ambion; antagomir two from Exiqon (miRCURY LNA microRNA Power Inhibitor; 4100464–002), and antagomir three from Thermo Scientific Dharmacon (miRIDIAN hairpin inhibitor; IH-300708–07). miR-335 Mimic oligonucleotide was obtained from Exiqon (473600–001). The following chemical reagents were used for cell treatment: Erlotinib Hydrochloride 99% from LGM Pharmaceutical Inc, pyridone 6 (P6) from Calbiochem, Trichlostatin (TSA), and 5-aza-2-deoxycytidine from Sigma-Aldrich.

## Antibodies

### For flow cytometry

APC anti-human AXL antibody (R and D Systems); cat # FAB154A.
Alexa Fluor 488 anti-human AXL (R and D Systems); cat # FAB154G.
PE-CF594 anti-human CD45 antibody (BD Biosciences); cat # 562279.
BV421 anti-human CD31 antibody (BD Biosciences); cat # 564089.
Alexa Fluor 488 anti-human CD326 (EpCAM) antibody (BioLegend); cat # 324210.

### For immunofluorescence

Alexa Fluor 488 Phalloidin (Thermo Fisher); cat # A12379.

### For immunoprecipitation and immunoblot analysis

AXL M-20 goat polyclonal IgG (SCBT); cat # sc-1097. Currently discontinued.
Anti-alpha-tubulin antibody (Millipore); cat # MABT205.
Phospho- Tyr PY20 mouse monoclonal IgG (SCBT); cat # sc-508.
Phospho-AXL mouse monoclonal IgG (R and D Biosystems); cat # MAB6965.
Anti-Gas6 Antibody (A-9): mouse monoclonal IgG (SCBT); cat # sc-376087.
GAPDH mouse monoclonal IgG (R and D Biosystems); cat # MAB5718.
Ras-GAP goat polyclonal IgG (R and D Biosystems); cat # AF5094.

## Additional information

### Funding

No external funding was received for this work.

## Author contributions

Polona Safaric Tepes, Data curation, Formal analysis, Investigation, Visualization, Methodology, Writing - review and editing; Debjani Pal, Conceptualization, Data curation, Formal analysis, Validation, Investigation, Visualization, Writing - review and editing; Trine Lindsted, Data curation, Formal analysis, Validation, Methodology; Ingrid Ibarra, Conceptualization, Data curation, Formal analysis, Supervision, Funding acquisition, Validation, Methodology, Writing - review and editing; Amaia Lujambio, Data curation, Formal analysis, Methodology; Vilma Jimenez Sabinina, Data curation, Formal analysis; Serif Senturk, Conceptualization, Data curation, Formal analysis, Supervision, Funding acquisition, Validation, Writing - review and editing; Madison Miller, Data curation, Formal analysis, Writing - review and editing; Navya Korimerla, Data curation, Writing - review and editing; Jiahao Huang, Data curation; Lawrence Glassman, Resources, Data curation; Paul Lee, Kevin Hyman, Michael Esposito, Gregory J Hannon, Resources; David Zeltsman, Raffaella Sordella, Conceptualization, Resources, Formal analysis, Supervision, Funding acquisition, Investigation, Visualization, Writing - original draft, Writing - review and editing

## Author ORCIDs

Polona Safaric Tepes (iD) https://orcid.org/0000-0002-5833-739X
Amaia Lujambio (iD) http://orcid.org/0000-0002-2798-1481
Raffaella Sordella (iD) https://orcid.org/0000-0001-9745-1227

## Ethics

Human subjects: The collection of human lung tissue samples and blood for this study was covered by Northwell Health/Cold Spring Harbor Laboratory IRB #TDP-TAP 1607 (Raffaella Sordella/10/11/16 ). The samples were acquired from patients already undergoing thoracic procedures (e.g. surgical tumor resection, biopsy) at Huntington Hospital. All study participants provided informed consent for the use of their lung tissue and blood for research purposes. Participants were informed of study aims, the potential risks and benefits of participation, and that any discoveries facilitated by the analysis of their tissues might be published. The participants were informed that their names would not be associated their samples in any publication or presentation of research findings.

## Decision letter and Author response

Decision letter https://doi.org/10.7554/eLife.66109.sa1
Author response https://doi.org/10.7554/eLife.66109.sa2

# Additional files

## Supplementary files

- Supplementary file 1. A complete list of primers for quantitative real-time PCR analysis of miRNAs
.
- Transparent reporting form

## Data availability

The data generated or analyzed during this study are included in the manuscript and supporting files.

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
