## [Decision Letter]

**Acceptance summary:**

This manuscripts demonstrate a potentially new paradigm in the evolution of cancer drug resistance by describing selection of a non-genetic pre-existing subpopulation. The authors show that AXL positive cells are pre-existing and AX^L+^ cells are enriched upon EGFR TKI resistance. In addition. they find that miR-355 expression is decreased in the resistant cells and AX^L+^ cells, that miR-355 regulates TGFbeta/ EMT signaling, CpG1 methylation level is increased in AX^L+^ve cells and this CpG island present in the promoter of MEST, which contains miR-335. Thus, this manuscript presents a new mechanism of AXL regulation and sheds some insights into non-genetic resistance.

**Decision letter after peer review:**

[Editors’ note: the authors submitted for reconsideration following the decision after peer review. What follows is the decision letter after the first round of review.]

Thank you for submitting your article "An epigenetic switch regulates the ontogeny of AXL positive/ EGFR-TKI resistant cells by modulating miR-335 expression" for consideration by *eLife*. Your article has been reviewed by 2 peer reviewers, and the evaluation has been overseen by Michael Green as the Reviewing Editor and Kevin Struhl as the Senior Editor. The following individual involved in review of your submission has agreed to reveal his identity: Narendra Wajapeyee (Reviewer #1).

The reviewers have discussed the reviews with one another and the Reviewing Editor has drafted this decision to help you prepare a revised submission.

Summary:

In this manuscript, Safaric Tepes, Pal et al. examine the role of AXL positive cell sub-populations in EGFR mutant lung cancer cell lines in response and resistance to EGFR TKIs. First, Safaric Tepes, Pal et al. examine the frequency of AXL positive cells in untreated lung cancer cell lines and human tumor tissue and show that all cases examined (including different oncogenic drivers) contain a similar AXL positive subpopulation of about 2-10%. These AXL positive cells exhibit mesenchymal features and are less sensitive to TKI. Interestingly, the AXL positive and negative state can introconvert over weeks, consistent with an epigenetic rather than genetic mechanism. To investigate how the AXL postive/negative state transition is regulated, Safaric Tepes, Pal et al. performed miRNA profiling of H1650 and H1650-M3 cells and identified miR-355 to be downregulated in the AXL-positive H1650-M3 cells. Inhibition of miR-355 using antagomirs increased the percentage of AXL positive cells and mesenchymal markers and decreased drug sensitivity, suggesting that miR-355 regulates the switch between AXL positive/negative states. They next examined the methylation status of the promoter of the MEST gene (which contains miR-355 in intron 2) and discovered that AXL positive cells uniquely contain a methylated CpG island. Decreasing methylation using decitabine lead to a decrease in AXL expression, suggesting that epigenetic manipulation of the AXL positive subpopulation might decrease the potential for EGFR mutant NSCLC cells to develop adaptive resistance to TKI.

Understanding how phenotypically distinct sub-populations may contribute to adaptive resistance to targeted therapies is very important for developing new therapeutic strategies that may prevent the development of acquired drug resistance. These studies follow prior studies by this group and others showing that non-genetic mechanisms of adaptive resistance such as EMT-like changes play a role in survival of cells early after initiation of treatment. While the demonstration of miR-355 as a regulator of the AXL positive state in sub-populations of EGFR mutant lung cancer cells is interesting, the novelty is somewhat muted by prior studies demonstrating that EMT-like clones pre-exist treatment (Bhang et al. Nature Medicine 2015) and that EMT-like resistance is regulated by miRNAs and methylation status (Shien et al. Cancer Research 2013). As discussed below, additional experiments addressing causality and specificity of the mir-355 mechanism would greatly strengthen the manuscript, along with directly demonstrating that modulating this pathway (e.g. decitabine) impacts the survival of persister cells and/or development of resistance.

In addition, this study makes a very concise point and may be more appropriate as a Short Report rather than as an Article for publication in *eLife*.

Essential revisions:

1. The manuscript focuses on miR-355 as a key regulator of the AXL-positive state, however it is difficult to interpret whether this miR-355 pathway is the central regulator of this phenotype switch, or just one of many. In the miRNA profiling experiment shown in Figure 4A-B, there were many miRNAs that were more up/down-regulated than miR-355. While the authors provide reasons why miR-355 was chosen for study, it is not clear why the other miR-355 were not examined as well. For instance, miR-200C was more down-regulated in the H1650-M3 cells than miR-355, and has been previously shown to be suppressed in mesenchymal gefitinib-resistant HCC827 cells (Shien, et al. " Acquired Resistance to EGFR Inhibitors Is Associated with a Manifestation of Stem cell-like Properties in Cancer Cells." Cancer Research 2013). Moreover, in this study, suppression of miR-200c was shown to be due to hypermethylation, which could be de-repressed by decitabine. To address this, first, the authors should discuss why other miRNAs identified in the profiling were not chosen for validation, and at least examine whether other miRNAs known to be relevant (such as miR-200c) could also be involved in this system. This is important because most of the experiments only show correlation between miR-355 suppression and AXL-positive/mesenchymal state, with the one experiment specifically testing causality (Figure 5C) showing only a modest increase in AXL positive cells after treatment with miR355 antagomir. Incidentally, the normalization used in Figure 5C is confusing – does this essentially mean that the absolute percentage of AXL-positive cells in the population went from approximately 10% to 15%? Second, in demonstrating that miR-355 suppression leads to increase in AXL-positive cells, mesenchymal genes and loss of drug sensitivity (Figure 5), it is noticeable that the authors chose to use different cell lines in the various sub-panels. For instance, H1650 and PC9 cells were used to show increase in AXL positive cells (Figure 5C) and upregulation of mesenchymal genes Figure 5E), however drug sensitivity was not shown for antagomir-treated PC9 cells (H4006 and HCC827 cells were shown instead). To demonstrate mesenchymal morphology after antagomir treatment, A549 cells were used instead of EGFR cell lines.

Why were different cell lines used for different experiments? It would be more convincing if the authors were able to show a consistent set of cell lines throughout all experiments.

2. The authors show that one of the CpG islands in the MEST promotor was differentially methylated in AXL positive cells, and that this could be reversed with decitabine. Similar to the comments above about specificity, decitabine undoubtedly has widespread effects on the expression of many genes and miRNAs. To demonstrate that the decrease in AXL mRNA expression after decitabine treatment is mediated by miR-355, the experiment in Figure 6H should be done in using an miR-355 antagomir to block the effects of decitabine. Also, miR-355 could be expressed in AXL-positive cells to see if this drives cells back into the AXL-negative state. A minor point: it is difficult to interpret how the AXL mRNA expression in Figure 6H translates into changes in the percentage of cells that are AXL-positive. It would be helpful to show the actual % AXL positive cells, which is mentioned in the text but not included in the figures.

3. The authors show that all cell lines and patient tumors tested contain approximately the same frequency of AXL-positive cells (2-10%). They also show that the AXL-positive cells have a mesenchymal phenotype. On the other hand, it is well known that different cancer cell lines exhibit phenotypes that range along the epithelial – mesenchymal spectrum. How do the authors reconcile this apparent discrepancy? Does a mesenchymal cell line (traditionally determined by looking at the bulk population) simply have a larger percentage of AXL-positive cells? Are AXL-positive cells a subset of mesenchymal cells, or a distinct but overlapping population? Some further clarification of the identity of the AXL-positive population would be helpful.

4. The most significant implication of this study is that the AXL-positive subpopulation may play a central role in adaptive resistance and ultimately acquired resistance to TKI, and that altering methylation status and inducing miR-335 expression may be a way of decreasing the AXL-positive population and preventing/delaying resistance. However, several questions are raised when interpreting this study in the context of prior studies. For instance, many groups have shown that the sensitivity of H1650 and H4006 cells to EGFR inhibitors is less than PC9 and HCC827 cells – yet this doesn't correlate well with the percentage of AXL-positive cells shown in Figure 1C. Also, the population of AXL-positive cells seems to be much higher than in prior studies looking at surviving PC9 drug tolerant persisters (Sharma et al. Cell 2010) or pre-existing EMT clones in HCC827 cells (Bhang et al. Nature Medicine 2015). To establish the relevance of these studies for adaptive/acquired resistance, I think it is very important for the authors to directly test whether modulation of the miR-355 pathway/AXL-positive sub-population impacts the development of resistance. First, are the pre-treatment AXL-positive cells the ones that preferentially persist during drug treatment? This could be easily determined by labeling sorted AXL positive and negative populations, recombining to reconstitute the proportions of the parental population, and treating with drug for a few weeks and analyzing the persistent population. Second, what is the effect of decitabine on AXL-positive cells sorted from non-treated parental cells? The effects of decitabine are tested only on H1650-M3 cells, however as the authors point out, these cells have a more permanent phenotype with less plasticity to convert back to AXL-negative cells – so it is not clear how well they accurately resemble the AXL-positive cells in the untreated cell population, which is the relevant cell population. Third, does decitabine treatment increase drug sensitivity of AXL-positive cells and decrease the number of persister cells that survive over time? This is a critical experiment to show that the mechanism uncovered here is relevant to the development of acquired resistance.

5. Does this mechanism of resistance occur in the context of other EGFR inhibitors or it is specific to erlotinib?

[Editors’ note: further revisions were suggested prior to acceptance, as described below.]

Thank you for resubmitting your work entitled "An epigenetic switch regulates the ontogeny of AXL positive/EGFR-TKI resistant cells by modulating miR-335 expression" for further consideration by *eLife*. Your revised article has been evaluated by Kevin Struhl as the Senior Editor, Michael Green as the Reviewing Editor, and two expert reviewers. The reviewers find your work of interest and appropriate for *eLife* but they have offered a number of suggestions for improvement, two of which are essential for you to address. We apologize for the long delay in getting this manuscript reviewed.

Summary:

The authors demonstrate a potentially new paradigm in the evolution of drug resistance by describing selection of a non-genetic pre existing subpopulation. This is an exciting new concept in oncology. Authors show that AXL positive cells are pre-existing and AX^L+^ cells are enriched upon EGFR TKI resistance. This data is interesting but lack some controls and and hard to put into context given the modest level of resistance they observe. Next, author identify that miR-355 expression is decreased in the resistant cells and AX^L+^ cells, that miR-355 regulates TGFbeta/ EMT signaling, CpG1 methylation level is increased in AX^L+^ve cells and this CpG island present in the promoter of MEST, which contains miR-335. The connection between this microRNA and AXL is novel and solid. The paper presents a new mechanism of AXL regulation and sheds some insights into non-genetic resistance. In light of the absence of patient or in vivo data, I think that the specific steps that AX^L+^ cells play in resistance still could be better fleshed out.

Essential revisions:

1. Is it possible that AX^L+^ cell enrichment in resistant cells is from the de novo formation of new AX^L+^ cells rather than selection from pre-existing AX^L+^ cells. Among other possible methods, lineage tracing studies in sorted AX^L+^ cells in the parental cell line then treated with Erlotinib could determine if truly they are the source of resistance. This is important to solidify the ontogeny of these cells.

2. Testing erlotinib sensitivity upon 1) miR-335 mimic transfection and 2) 5-Aza-dC treatment in PC14 and H1650-M3 cells rather than H1650 cells would further strengthen the possible role of miR-335 and MEST promoter methylation in Erlotinib resistance.

Strongly encouraged revisions:

1. Authors described that PC14 was generated from PC9 upon Erlotinib-selection (p6, line 159). In many cases that have been previously published, resistant cells have at least 100 fold higher IC50 compared to the parent cells. Often such cells are carried/passaged in 1uM EGFR TKI, indicating true EGFR independence. In contrast these cells seem barely resistant (Figure 2A). Furthermore if AXL was the driver of resistance in these cells shouldn’t the ic50 to erlotinib by higher than 3nM? It appears that these cells are then still EGFR dependent.

2. The data would be stronger with more accessible/clear models of resistance to osimertinib (such as PC9 and H1975 cells) which form rapidly, and are more clinically relevant. Do AX^L+^ cells pre-exist there and are they enriched after treatment?

3. IC50 difference between PC9-AXL-ve and PC9-AX^L+^ve cells is not that dramatic, 1.4->4.4nM (Figure 2). As authors stated in p15 line 427, the slight increase in IC50 in AX^L+^ cells could have been resulted from slower proliferation rate and not directly associated with Erlotinib resistance? It would be better to show this multiple ways such as colony formation assays.

4. Drug sensitivity in Figure 2—figure supplement 1 seems to be very different from Figure 2 data. Here cells were tested with 0.5 μm erlotinib, but still more than 30% viable cells present in this experiment. Please specify how the data was normalized and compared.

5. Authors found that parent cells try to maintain the percentage of negative and positive population (Figure 3). Is this also driven by change in methylation status and miR-335? Measuring relative miR-335 expression levels and methylation level at the end of the time point when cells reach equilibrium would be preferred.

---

## [Author Response]

[Editors’ note: the authors resubmitted a revised version of the paper for consideration. What follows is the authors’ response to the first round of review.]

Essential revisions:1. The manuscript focuses on miR-355 as a key regulator of the AXL-positive state, however it is difficult to interpret whether this miR-355 pathway is the central regulator of this phenotype switch, or just one of many. In the miRNA profiling experiment shown in Figure 4A-B, there were many miRNAs that were more up/down-regulated than miR-355. While the authors provide reasons why miR-355 was chosen for study, it is not clear why the other miR-355 were not examined as well. For instance, miR-200C was more down-regulated in the H1650-M3 cells than miR-355, and has been previously shown to be suppressed in mesenchymal gefitinib-resistant HCC827 cells (Shien, et al. " Acquired Resistance to EGFR Inhibitors Is Associated with a Manifestation of Stem cell-like Properties in Cancer Cells." Cancer Research 2013). Moreover, in this study, suppression of miR-200c was shown to be due to hypermethylation, which could be de-repressed by decitabine. To address this, first, the authors should discuss why other miRNAs identified in the profiling were not chosen for validation, and at least examine whether other miRNAs known to be relevant (such as miR-200c) could also be involved in this system.

Our miRNA expression profile identified 20 miRNAs that were up-regulated and 19 miRNAs that were down-regulated in the AXL-positive cells (H1650-M3) compared to the AXL-negative cells (H1650). These were independently validated by quantitative stem-loop RT-PCR. Although none of the identified miRNAs were predicted to target AXL, we were intrigued by the differential expression of miR-335 we observed in AXL-positive compared to AXL-negative cells. In fact, miR-335 has been reported to suppress a mesenchymal-like state and metastatic dissemination by targeting a diverse set of genes regulating cell migration, extra-cellular matrix remodeling, cell selfrenewal and epigenetic reprograming (Tavazoie, Alarcon et al. 2008) (Figure 4D). Among them, of particular interest was the regulation of the TGF-_β_eta_β_axis by miR-335. In fact, TGF-_ββ_is a well known regulator of AXL and AXL activity (Lynch, Fay et al. 2012). Furthermore, the TGF-_β_eta_β_axis has also been shown to suppress the expression of multiple miRNAs that we found to be down-regulated in AXL-positive cells (Gregory, Bracken et al. 2011, Yang, Li et al. 2012, Kato, Dang et al. 2013). Altogether these findings let us to hypothesize that differentially expressed miRNAs in AXL-positive cells could be part of a hierarchically organized miRNA cluster primed by miR-335 and that the regulation of miR-335 could play a major role in the ontogeny of AXLpositive cells. As described in the manuscript we have verified that miR-335 regulate the expression of TGF-β signaling and that the later is sufficient to modulate the expression of the majority of miRNAs we found to be deregulated in AXL-ve cells except for miR-335 (Figure 4H-J).

This is important because most of the experiments only show correlation between miR-355 suppression and AXL-positive/mesenchymal state, with the one experiment specifically testing causality (Figure 5C) showing only a modest increase in AXL positive cells after treatment with miR355 antagomir. Incidentally, the normalization used in Figure 5C is confusing – does this essentially mean that the absolute percentage of AXL-positive cells in the population went from approximately 10% to 15%?

To better establish the role of miR-335 in restricting the generation of AXL-negative cells, we have tested causality not only in the context of AXL expression (Figure 5A,B) but also in regards to the acquisition of other molecular and phenotypic features that characterize the AXL-negative cells such as expression of mesenchymal markers (Figure 5D) and signature miRNAs (Figure 4J) as well as Erlotinib sensitivity (Figure 5E and F) and morphological features (Figure 5C). We have also updated figure 5A and validated the role of miR-335 using a CRISPR based approach (Figure 5—figure supplement 2.

Second, in demonstrating that miR-355 suppression leads to increase in AXL-positive cells, mesenchymal genes and loss of drug sensitivity (Figure 5), it is noticeable that the authors chose to use different cell lines in the various sub-panels. For instance, H1650 and PC9 cells were used to show increase in AXL positive cells (Figure 5C) and upregulation of mesenchymal genes Figure 5E), however drug sensitivity was not shown for antagomir-treated PC9 cells (H4006 and HCC827 cells were shown instead). To demonstrate mesenchymal morphology after antagomir treatment, A549 cells were used instead of EGFR cell lines. Why were different cell lines used for different experiments? It would be more convincing if the authors were able to show a consistent set of cell lines throughout all experiments.

All the experiments described in Figure 5 have now been performed using the same panel of cell lines.

2. The authors show that one of the CpG islands in the MEST promotor was differentially methylated in AXL positive cells, and that this could be reversed with decitabine. Similar to the comments above about specificity, decitabine undoubtedly has widespread effects on the expression of many genes and miRNAs. To demonstrate that the decrease in AXL mRNA expression after decitabine treatment is mediated by miR-355, the experiment in Figure 6H should be done in using an miR-355 antagomir to block the effects of decitabine.

This is an excellent point. To address this, we transfected H1650 and H1650-M3 cells with AntagomiR-335, and treated the cells in the presence or absence of 5-Azacytidine (Decitabine) at different concentrations. Consistent with our hypothesis, antagomiR-335 transfection even in the presence of 5-Azacytidine was sufficient to increase the number of AXL-positive cells (Figure 6G) and to restore the expression of AXL to levels comparable to the one observed in AXL-positive cells (Figure 6—figure supplement 1 C).

Also, miR-355 could be expressed in AXL-positive cells to see if this drives cells back into the AXL-negative state.

Unfortunately when we transfected Axl +ve cells (H1650-M3, PC14) with miR-335 mimic, the cells died precluding the possibility to perform such experiment.

A minor point: it is difficult to interpret how the AXL mRNA expression in Figure 6H translates into changes in the percentage of cells that are AXL-positive. It would be helpful to show the actual % AXL positive cells, which is mentioned in the text but not included in the figures.

We have modified the chart to show % AXL positive cells accordingly.

3. The authors show that all cell lines and patient tumors tested contain approximately the same frequency of AXL-positive cells (2-10%). They also show that the AXL-positive cells have a mesenchymal phenotype. On the other hand, it is well known that different cancer cell lines exhibit phenotypes that range along the epithelial – mesenchymal spectrum. How do the authors reconcile this apparent discrepancy? Does a mesenchymal cell line (traditionally determined by looking at the bulk population) simply have a larger percentage of AXL-positive cells? Are AXL-positive cells a subset of mesenchymal cells, or a distinct but overlapping population? Some further clarification of the identity of the AXL-positive population would be helpful.

Epithelial-to-mesenchymal transition can be induced by multiple cues, including the over-expression of certain receptor tyrosine kinase receptors like AXL, c-MET, PDGFR; exposure to TGF-_β_1, TGF-_β_2; or hypoxia (Yao, Fenoglio et al. 2010, Wu, Hou et al. 2013, Zhang, Huang et al. 2013, Rankin, Fuh et al. 2014, Elkabets, Pazarentzos et al. 2015, Li, Dobbins et al. 2015). Hence, we wondered whether the expression of AXL was a common feature of all mesenchymal cells or if on the contrary was specific to a particular cell state. Hence, we analyzed the presence of AXL-positive cells in multiple tumor derived cell lines and correlate their distribution with the mesenchymal status of the cells. Despite H1703, H1975 and H23 cells present with clear mesenchymal characteristics, AXL-positive cells were virtually absent in these cell lines (Figure 2—figure supplement 2A-C). Hence we concluded that while all mesenchymal cells share common characteristics such as increased stress fibers, increased motility, elongated shape, etc.; AXL-positive cells are a unique cell population with features that only partially overlap with other mesenchymal cells.

4. The most significant implication of this study is that the AXL-positive subpopulation may play a central role in adaptive resistance and ultimately acquired resistance to TKI, and that altering methylation status and inducing miR-335 expression may be a way of decreasing the AXL-positive population and preventing/delaying resistance. However, several questions are raised when interpreting this study in the context of prior studies. For instance, many groups have shown that the sensitivity of H1650 and H4006 cells to EGFR inhibitors is less than PC9 and HCC827 cells – yet this doesn't correlate well with the percentage of AXL-positive cells shown in Figure 1C. Also, the population of AXL-positive cells seems to be much higher than in prior studies looking at surviving PC9 drug tolerant persisters (Sharma et al. Cell 2010) or pre-existing EMT clones in HCC827 cells (Bhang et al. Nature Medicine 2015). To establish the relevance of these studies for adaptive/acquired resistance, I think it is very important for the authors to directly test whether modulation of the miR-355 pathway/AXL-positive sub-population impacts the development of resistance.First, are the pre-treatment AXL-positive cells the ones that preferentially persist during drug treatment? This could be easily determined by labeling sorted AXL positive and negative populations, recombining to reconstitute the proportions of the parental population, and treating with drug for a few weeks and analyzing the persistent population.

As suggested, we sorted Axl +ve and Axl –ve cells from the PC9 cell line, we infected PC9 Axl +ve cells with a lentivirus expressing Td-Tomato (Td-Tom), mixed PC9-Td-Tom Axl +ve cells with PC9-Axl –ve cells in different ratios and treated them with Erlotinib (Figure 2—figure supplement 1 A-B). As we increased the proportion of Axl +ve cells, the number of cells that survived the drug treatment comparably increased.

Interestingly, when we then analyzed the representations of AX^L+^ve cells in the Td-Tom negative persistent “persistent” cell population, we observed that 40% of the persistent cells were AX^L+^ve.

Second, what is the effect of decitabine on AXL-positive cells sorted from non-treated parental cells? The effects of decitabine are tested only on H1650-M3 cells, however as the authors point out, these cells have a more permanent phenotype with less plasticity to convert back to AXL-negative cells – so it is not clear how well they accurately resemble the AXL-positive cells in the untreated cell population, which is the relevant cell population.

As suggested we have expanded our studies to test the effect of decitabine on AXL-positive cells FACS-sorted from non-treated parental cells (Figure 6—figure supplement 1 D). Similar to the case of H1650-M3 cells (Figure 6—figure supplement 1 E), we observed that the % of Axl +ve cells significantly decreased upon treatment with 5-Aza.

Third, does decitabine treatment increase drug sensitivity of AXL-positive cells and decrease the number of persister cells that survive over time? This is a critical experiment to show that the mechanism uncovered here is relevant to the development of acquired resistance.

As shown in figure 6 H, we observed that 5-Aza treatment increased the sensitivity of AX^L+^ve cells to levels comparable to the one observed in AXL-ve cells (Figure 6 H).

5. Does this mechanism of resistance occur in the context of other EGFR inhibitors or it is specific to erlotinib?

As shown in Figure 2—figure supplement 1 A-B we didn’t observe any major differences when we treated cells with two different EGFR TKi.

[Editors’ note: what follows is the authors’ response to the second round of review.]

Essential revisions:1. Is it possible that AX^L+^ cell enrichment in resistant cells is from the de novo formation of new AX^L+^ cells rather than selection from pre-existing AX^L+^ cells. Among other possible methods, lineage tracing studies in sorted AX^L+^ cells in the parental cell line then treated with Erlotinib could determine if truly they are the source of resistance. This is important to solidify the ontogeny of these cells.

To address this point we performed the following experiment:

– First, we stably infected PC9 and H1650 with a lentiviral vector expressing GFP. Then we sorted AX^L+^ /GFP+ and AXL-/GFP- cells from the PC9 and H1650 cell lines. AX^L+^ /GFP+ cells and AXL-/GFP- cells were immediately mixed at a 1:1 ratio and plated in 96 well plates.

– The next day we confirmed the approximate 50% representation of GFP+/GFP- cells by FACS and started the Erlotinib treatment.

– After 72h the % of GFP+ cells was determined by Guava Flow Cytometer. A maximum of 6000 cells was analyzed.

As shown in Figure 2—figure supplement 1, we observed an increased representation of GFP+ cells upon Erlotinib treatment. Given the experiment was conducted within 96 hours upon sorting, unlikely the GFP+ Erlotinib resistant cells could have been generated by a de novo mechanism.

2. Testing erlotinib sensitivity upon 1) miR-335 mimic transfection and 2) 5-Aza-dC treatment in PC14 and H1650-M3 cells rather than H1650 cells would further strengthen the possible role of miR-335 and MEST promoter methylation in Erlotinib resistance.

Unlike the case of AXL- cells (H1650 and PC9), when we transfected the AX^L+^ cells H1650-M3 and PC14 we observed a dramatic decreased in the number of viable cells during time compare to control (Figure 5 F and Figure5-Supplementary figure 3). To exclude this was due to a non-specific effect, the experiment was repeated by transfecting an unrelated miRNA (miR-183).

Consistent with the observation that miR-355 is regulated by DNA methylation, we observed that AX^L+^ cells (H1650-M3 and PC14) are also more sensitive to long treatment with 5-Aza-dC (Figure 6-Supplementary figure 1). Of note, from a clinical perspective, this result is of particular interest given the sensitivity of the AX^L+^ cells is within the range of blood levels of Decitabine observed in clinical trials.

Strongly encouraged revisions:1. Authors described that PC14 was generated from PC9 upon Erlotinib-selection (p6, line 159). In many cases that have been previously published, resistant cells have at least 100 fold higher IC50 compared to the parent cells. Often such cells are carried/passaged in 1uM EGFR TKI, indicating true EGFR independence. In contrast these cells seem barely resistant (Figure 2A). Furthermore if AXL was the driver of resistance in these cells shouldn’t the ic50 to erlotinib by higher than 3nM? It appears that these cells are then still EGFR dependent.

As the reviewer indicated, in the past, it has been reported a lower Erlotinib IC50 in PC14 cells. In our experience, differences in growing conditions (i.e., cell confluence, amount of serum) as well methods to measure the number of resistant cells can lead to differences in IC50. In particular assessing the number of cells based on differences in the cell metabolism, such as in the case of the MTT assay, could result in greater differences in IC50.

2. The data would be stronger with more accessible/clear models of resistance to osimertinib (such as PC9 and H1975 cells) which form rapidly, and are more clinically relevant. Do AX^L+^ cells pre-exist there and are they enriched after treatment?

Osimertinib is a third-generation EGFR TKi that has been recently approved as first-line treatment for NSCLC harboring EGFR mutations. Although it has been observed that Osimertinib has an increased clinical efficacy when compared to Erlotinib or Gefitinib, the last two are still the most commonly used EGFR Tki in the clinic. Hence, we believe our studies are clinically relevant.

Interestingly, Taniguchi et al. have recently shown that treatment with Osimertinib in PC9 as well as in other NSCLC cells resulted in the occurrence of Erlotinib resistant/AX^L+^ cells(Taniguchi, Yamada et al. 2019). The observation that, as shown in Figure 2—figure supplement 2, AX^L+^ cells are already present in PC9 and H1975 cells before treatment, supports the idea that also in these model systems EGFR TKi resistant cells are pre-existing.

3. IC50 difference between PC9-AXL-ve and PC9-AX^L+^ve cells is not that dramatic, 1.4->4.4nM (Figure 2). As authors stated in p15 line 427, the slight increase in IC50 in AX^L+^ cells could have been resulted from slower proliferation rate and not directly associated with Erlotinib resistance? It would be better to show this multiple ways such as colony formation assays.

To address this point we performed the following experiment:

– We stably infected PC9 Axl +ve cells with lentivirus expressing Td-Tomato (Td-Tom). Then we mixed PC9-Td-Tom Axl +ve cells with PC9-Axl –ve cells in different ratios as following:

W10: 100% PC9-Axl –ve cells

R1W9: 10% PC9-Td-Tom Axl +ve cells + 90% PC9-Axl –ve cells

R5W5: 50% PC9-Td-Tom Axl +ve cells + 50% PC9-Axl –ve cells

R9W1: 90% PC9-Td-Tom Axl +ve cells + 10% PC9-Axl –ve cells

R10: 100% PC9-Td-Tom Axl +ve cells

– We then plated 20,000 cells of each group (W10, R1W9, R5W5, R9W1, R10) in multiple 6 wells, and treated them with Erlotinib at a concentration of 100x the IC50 concentrations for 2 weeks. We then removed the drugs and let the persisters cells grow for 1 more week. After that, we counted the number of surviving colonies and the number of colonies with more than 100 cells.

– We found that the initial representation of Axl +ve cells correlates with the number of colonies we obtained after treatment. Of note, subsequent analysis of the colonies indicates in the W10 group, 45% of the cells were in an AXL +ve state. See Author response image 1.

**Author response image 1. respfig1:** Pre-existent Axl +ve cells preferentially persist during drug treatment. The charts depict the percentage of surviving colonies with more than 100 cells after 15 days of Erlotinib treatment, normalized to R10. Each bar represents mean ± SD of 3 replicates from two independent experiments. n.s. = non-significant. (p-value * < 0.05, * *< 0.005, unpaired t-test). W10: 100% PC9-Axl –ve cells; R1W9: 10% PC9-Td-Tom Axl +ve cells + 90% PC9-Axl –ve cells; R5W5: 50% PC9-Td-Tom Axl +ve cells + 50% PC9-Axl –ve cells; R9W1: 90% PC9-Td-Tom Axl +ve cells + 10% PC9-Axl –ve cells;R10: 100% PC9-Td-Tom Axl +ve cells.

4. Drug sensitivity in Figure 2—figure supplement 1 seems to be very different from Figure 2 data. Here cells were tested with 0.5 μm erlotinib, but still more than 30% viable cells present in this experiment. Please specify how the data was normalized and compared.

Biological differences among experiments are expected due to differences in growing conditions (i.e., cell confluence, amount of serum) as well methods and times to measure the number of viable cells which leads to differences in drug sensitivity. Yet, these differences do not change the overall conclusion of the experiment.

5. Authors found that parent cells try to maintain the percentage of negative and positive population (Figure 3). Is this also driven by change in methylation status and miR-335? Measuring relative miR-335 expression levels and methylation level at the end of the time point when cells reach equilibrium would be preferred.

We apologize to the reviewer, but given the time frame of the experiment, we decided not to address this point.